# Population dynamics of immunological synapse formation induced by bispecific T cell engagers predict clinical pharmacodynamics and treatment resistance

**Can Liu[1], Jiawei Zhou[1], Stephan Kudlacek[2], Timothy Qi[1], Tyler Dunlap[1], Yanguang Cao[1,3]***

[1]Division of Pharmacotherapy and Experimental Therapeutics, School of Pharmacy, University of North Carolina at Chapel Hill, Chapel Hill, United States; [2]Department of Biochemistry and Biophysics, University of North Carolina at Chapel Hill, Chapel Hill, United States; [3]Lineberger Comprehensive Cancer Center, School of Medicine, University of North Carolina at Chapel Hill, Chapel Hill, United States

*For correspondence: yanguang@unc.edu

Competing interest: The authors declare that no competing interests exist.

**Abstract** Effector T cells need to form immunological synapses (IS) with recognized target cells to elicit cytolytic effects. Facilitating IS formation is the principal pharmacological action of most T cell-based cancer immunotherapies. However, the dynamics of IS formation at the cell population level, the primary driver of the pharmacodynamics of many cancer immunotherapies, remains poorly defined. Using classic immunotherapy CD3/CD19 bispecific T cell engager (BiTE) as our model system, we integrate experimental and theoretical approaches to investigate the population dynamics of IS formation and their relevance to clinical pharmacodynamics and treatment resistance. Our models produce experimentally consistent predictions when defining IS formation as a series of spatiotemporally coordinated events driven by molecular and cellular interactions. The models predict tumor-killing pharmacodynamics in patients and reveal trajectories of tumor evolution across anatomical sites under BiTE immunotherapy. Our models highlight the bone marrow as a potential sanctuary site permitting tumor evolution and antigen escape. The models also suggest that optimal dosing regimens are a function of tumor growth, CD19 expression, and patient T cell abundance, which confer adequate tumor control with reduced disease evolution. This work has implications for developing more effective T cell-based cancer immunotherapies.

## Editor's evaluation

This work provides an important finding, that aspects of clinical outcomes can be predicted by a random search to an immunological synapse-based computational model for T cells directed by specific engagers. It provides solid evidence based on in vitro synapse formation measurements using imaging flow cytometry. The work will be of interest to investigators in the still-expanding immunotherapy field, and also as an example of how biologic drugs interface with endogenous cellular resources in a patient.'

## Introduction

Immunotherapy is a type of cancer treatment that helps patients' immune systems fight cancer, and these therapies include immune checkpoint inhibitors and BiTE. Despite many successes, multiple challenges persist. For instance, only a fraction of patients respond, and many responders eventually relapse (*Nagorsen et al., 2012*; *Thakur et al., 2018*; *Topp et al., 2014*). The primary pharmacological action of cancer immunotherapies is to activate or reinvigorate the effector T cells to find and engage tumor cells and eventually form cytolytic IS. BiTE, as a unique type of cancer immunotherapy, can redirect T cells to bind specific antigens on tumor cells and form IS. Facilitating the formation of IS through tight intercellular apposition lead to cytolysis of the tumor cell. IS formation is, therefore, a critical step for BiTE pharmacodynamics (*Nagorsen et al., 2012*). Like the IS formed between antigen-presenting cells and T cells, BiTE-induced IS formation between tumor cells and T cells is a precisely orchestrated cascade of molecular and cellular interactions (*Delon and Germain, 2000*; *Roda-Navarro and Álvarez-Vallina, 2019*). Understanding the dynamics of IS formation at the cell population level and factors governing this process hold promise for predicting pharmacodynamics and treatment resistance, one of the most critical issues for immunotherapies.

The molecular mechanisms of IS formation have been widely studied to identify potential molecular targets for cancer immunotherapy (*Finetti and Baldari, 2018*; *Xiong et al., 2021*). However, IS formation at the macroscopic cell population level extends beyond molecular crosslinking and involves multiple steps of intercellular interactions, such as T cells scanning of target cells in the tumor microenvironment, slowing their motility upon recognition, and establishing intercellular adhesion in response to signals generated by the first encounter (*Dustin and Cooper, 2000*; *Fousek et al., 2021*). Previous pharmacodynamic models of BiTE immunotherapy have focused, almost exclusively, on the mechanisms of molecular crosslinking, with little consideration for macroscopic intercellular interactions (*Betts et al., 2019*; *Jiang et al., 2018*; *Schropp et al., 2019*; *Song et al., 2021*). We sought to investigate how intercellular interactions at the population level could provide further insight into IS formation dynamics, BiTE pharmacodynamics, and mechanisms of resistance.

Tumor cells reside in dynamic and heterogeneous microenvironments. Suboptimal BiTE efficacy could result from insufficient numbers of effector cells, poor drug penetration into tumor tissue, and antigen loss leading to immune escape. Characterizing IS formation dynamics under diverse conditions, such as varying T cell density, BiTE concentration, antigen binding affinity, and antigen expression, could provide insights into factors determining BiTE pharmacodynamics and possible mechanisms of tumor resistance. *Bell, 1978* developed theoretical models of cell-cell adhesionthat highlighted not only the importance of cell-surface receptors and ligands, but also biophysical factors including receptor diffusivity on the cell membrane and hydrodynamic shearing forces. By now, to our knowledge, there has been no theoretical framework developed to characterize the population dynamics of IS formation, limiting our ability to predict the pharmacodynamics of and resistance to cancer immunotherapies.

In this study, we applied imaging flow cytometry to quantify BiTE-induced IS formation dynamics under various experimental conditions. Additionally, we developed theoretical models to simulate IS formation on different spatiotemporal scales to interrogate factors influential to this process. After considering patient-specific parameters, models incorporating IS formation dynamics adequately predicted tumor-killing pharmacodynamics in patients. These models revealed trajectories of antigen escape and tumor evolution across anatomical sites and predicted optimal doses and regimens that could confer effective tumor control with reduced disease evolution. Our work shows substantial implications for developing effective T cell-based cancer immunotherapies.

## Results

### Experimental design and theoretical models

The formation dynamics of the BiTE-induced IS were the focus of this study. As depicted in *Figure 1a*, CD3$^+$ Jurkat was used as effector cells (E) while CD19$^+$ Raji was used as target cells (T). Jurkat and Raji cells were sorted into three subpopulations, high (H), medium (M), and low (L), based on their membrane expressions of CD3 or CD19, respectively (*Figure 1—figure supplement 1*). Effector and target cells were then co-incubated in the presence of blinatumomab. IS formation dynamics were visualized and quantified by imaging flow cytometry (*Figure 1a*, *Figure 1—figure supplement 2*). We

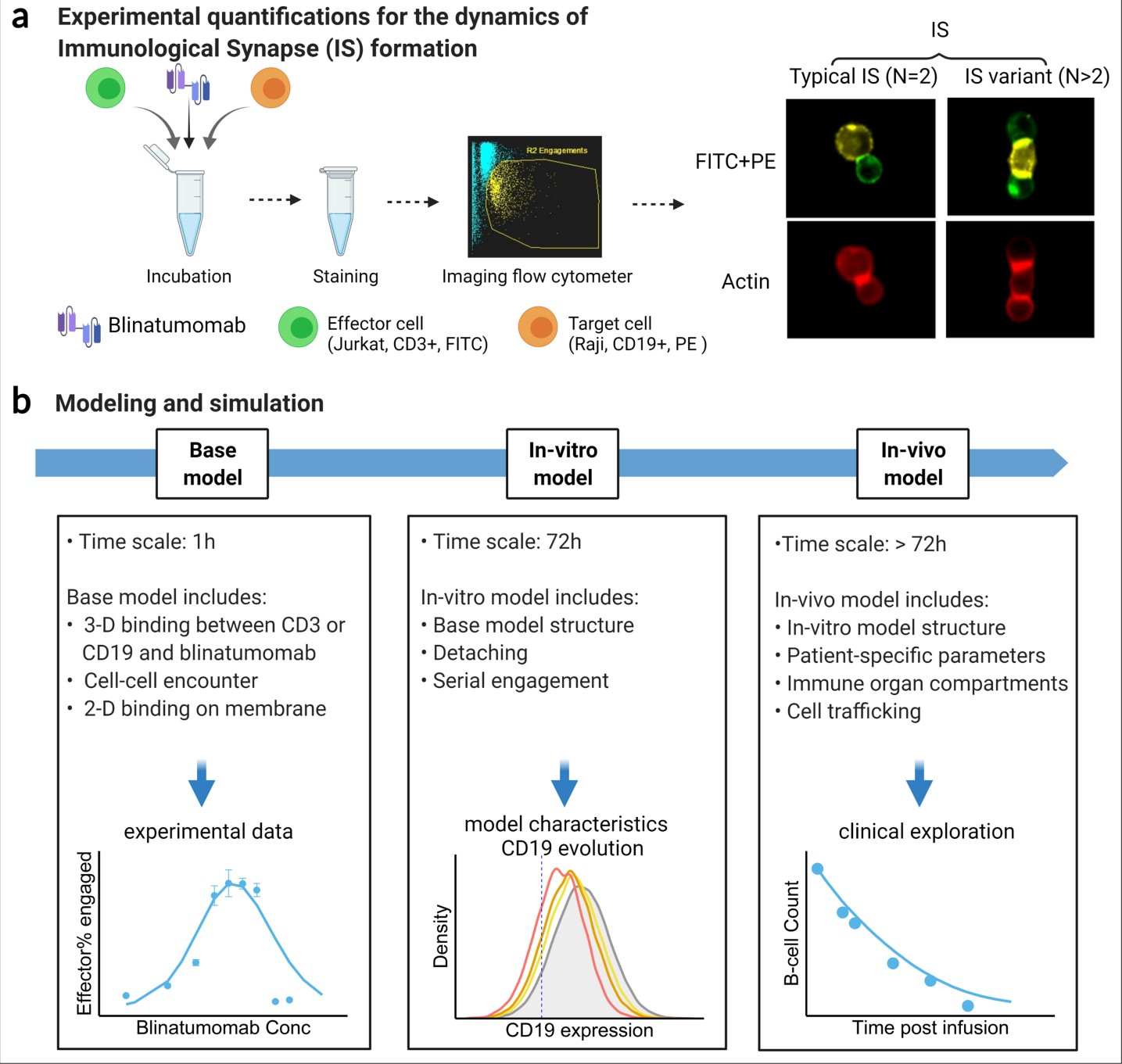

**Figure 1.** Schematic of experimental design and theoretical models. The study examined the formation dynamics of immunological synapses (IS) elicited by a bispecific T cell engager (BiTE). (**a**) The abundance and dynamics of IS formation were quantified by imaging flow cytometry under various experimental conditions. (**b**) Three mechanistic agent-based models were developed for the comprehensive characterization of cell-cell engagement and tumor-killing effects on different spatiotemporal scales. 3-D, three-dimensional; 2-D, two-dimensional.

The online version of this article includes the following figure supplement(s) for figure 1:

**Figure supplement 1.** Histogram and quantification of high (H), medium (M), and low (L) antigen expression (CD19 and CD3) in cell subpopulations.

**Figure supplement 2.** Image-based algorithms and gating strategy to identify typical immunological synapse (IS).

quantified the dynamics of IS formation under various experimental conditions, including different cell densities, antigen expressions, incubation durations, E:T ratios, and antibody concentrations. Multiple types of IS were observed and quantified, including 'typical' IS with one effector and one target cell (ET) and 'variants' with three or more cells engaged, such as ETE, ETT, and ETET.

We developed three mechanistic agent-based models to investigate IS formation and tumor-killing on different spatiotemporal scales. These models were developed in a stepwise fashion and calibrated with experimental or clinical data. As shown in *Figure 1b*, the base model was developed for predicting IS formation dynamics within 1 hr. The base model consists of three fundamental components during IS formation: three-dimensional (3D) binding between antibodies and antigens (CD3 or CD19) to form binary complexes, the probability of cell-cell encounter, and cell-cell adhesion driven by two-dimensional (2D) binding to form ternary complexes (CD3-BiTE-CD19) on the cellular membrane. The model structure for the base model is provided in the next section.

Next, the base model was extended with serial cell-cell engagement dynamics to capture IS formation and tumor-killing for up to 72 hr; we refer to this as the in vitro model. The in vitro model could evaluate the chance for target antigen loss (i.e. CD19), a common mechanism of immune escape, and therapeutic resistance to BiTEs (*Thakur et al., 2018*; *Topp et al., 2014*).

Last, the in vitro model was expanded to include cell-cell engagement in anatomically distinct compartments of the body. This model considered infiltration gradients of T and B cells and organ-to-organ cell trafficking. We refer to this expanded model as the in vivo model. This model integrated patient-specific parameters to predict clinically observed profiles of tumor killing and relapse. The in vivo model was also applied to support the simulation of antigen escape and tumor evolution across anatomical sites and compare dosing regimens for effective tumor control in light of tumor evolution.

## Model structure (Base model)

The major mechanism of BiTE pharmacology is to produce ternary complexes (CD3-BiTE-CD19) on opposing cell surfaces, driving IS formation. The dynamics of IS formation are, essentially, two independent and indispensable cellular processes mediated by the cell-cell encounter and cell-cell adhesion. Compared to molecular scale-focused pharmacodynamics models of BiTEs (*Betts et al., 2019*; *Jiang et al., 2018*; *Schropp et al., 2019*; *Song et al., 2021*), our models highlight the importance of these two cellular processes for IS formation in the context of macroscopic and biophysical forces (*Figure 2*).

In the base model (*Figure 2*), three essential steps are defined: antibody-antigen binding to form binary complexes (BiTE-CD3 and BiTE-CD19) (step 1), effector-target cell encounter probability defined as a function of cell mobility and density (step 2), and effector-target cell adhesion probability defined as a function of ternary complexes formed during contact (step 3). Model equations are provided in Appendix 1. Antibody-antigen binding to form binary complexes were assumed to be 3D processes that reached rapid equilibrium prior to cellular-scale events. Free effector cells were assumed to follow Brownian motion (*Celli et al., 2012*) and encounter target cells at a probability proportional to cell density and motility in the incubation environment (*Figure 2*).

After the cell-cell encounter, the probability of adhesion was modeled as a function of the number of ternary complexes (CD3-BiTE-CD19) formed between cells during the duration of contact. Cell-cell complexes with a high number of ternary complexes would, therefore, have a higher probability to adhering and eventually forming IS. Ternary complex formation on opposing cells was assumed to be restricted to the cell membrane (2D binding; *Appendix 1—figure 2*). The derivation of 2D binding affinities is provided in Appendix 1.5. Cells that failed to adhere upon encounter (futile contact) were assumed to diffuse away and become free to repeat the same process. Two cells engaged in a typical IS were also considered capable of engaging additional effector or target cells to become an IS variant, per our experimental observations (*Figure 2*). The model considered IS variants comprising up to four cells.

Model details and parameters for the base, in vitro and in vivo models are provided in materials and methods, Appendix 1, and *Supplementary file 1*.

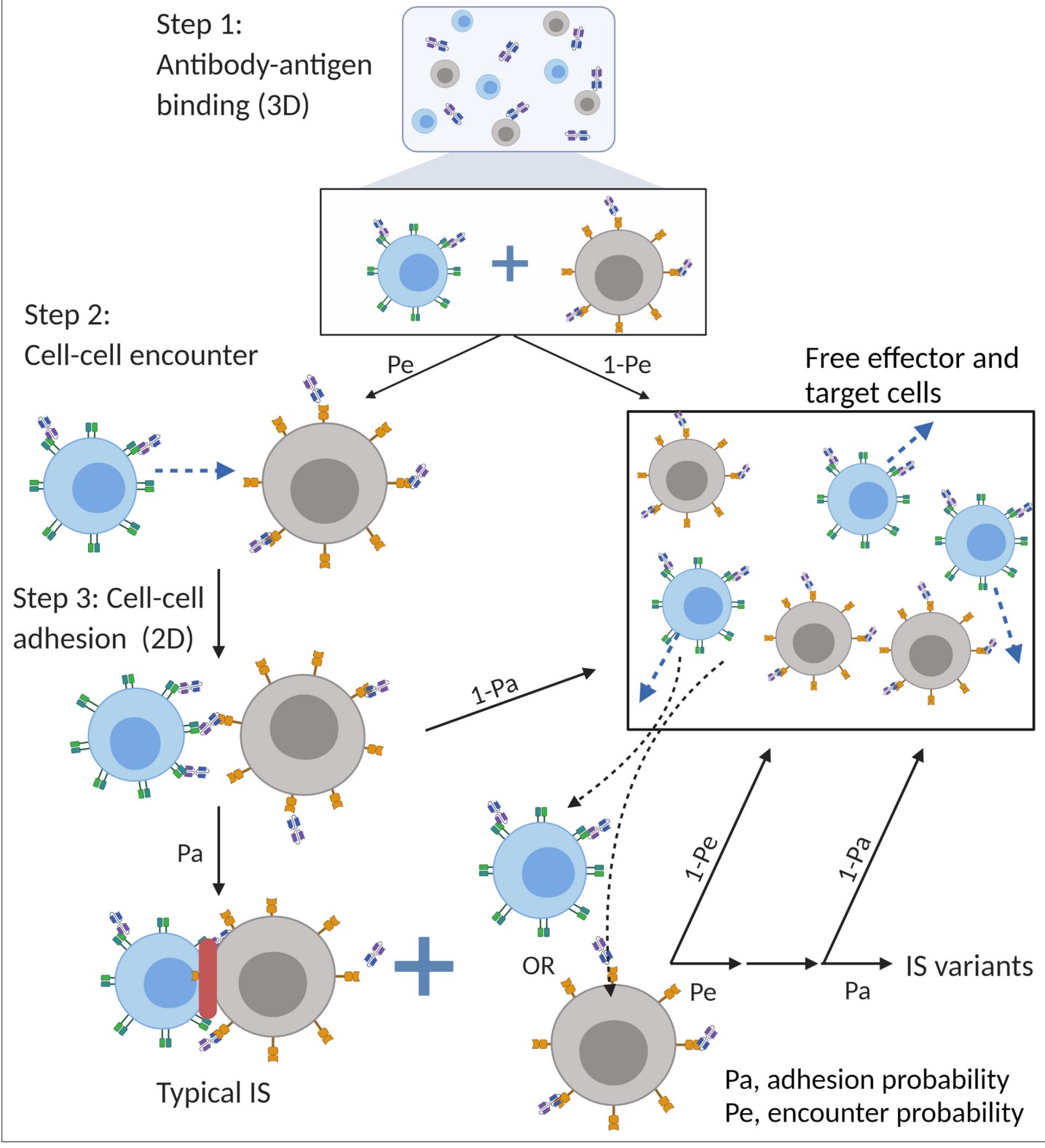

**Figure 2.** The base model included three essential steps to describe the process of immunological synapses (IS) formation induced by bispecific T cell engagers (BiTEs). Step 1: three-dimensional (3D) antibody-antigen binding in the media to form a binary complex; Step 2: cell-cell encounter, with encounter probability (Pe) dictated by cell motility and density; Step 3: cell-cell adhesion and IS formation, with adhesion probability (Pa) driven by the density of ternary complexes formed on the cell-cell contact area (two-dimensional (2D) binding) during contact. Newly formed typical IS had a chance to engage additional free effector or target cells to form an IS variant.

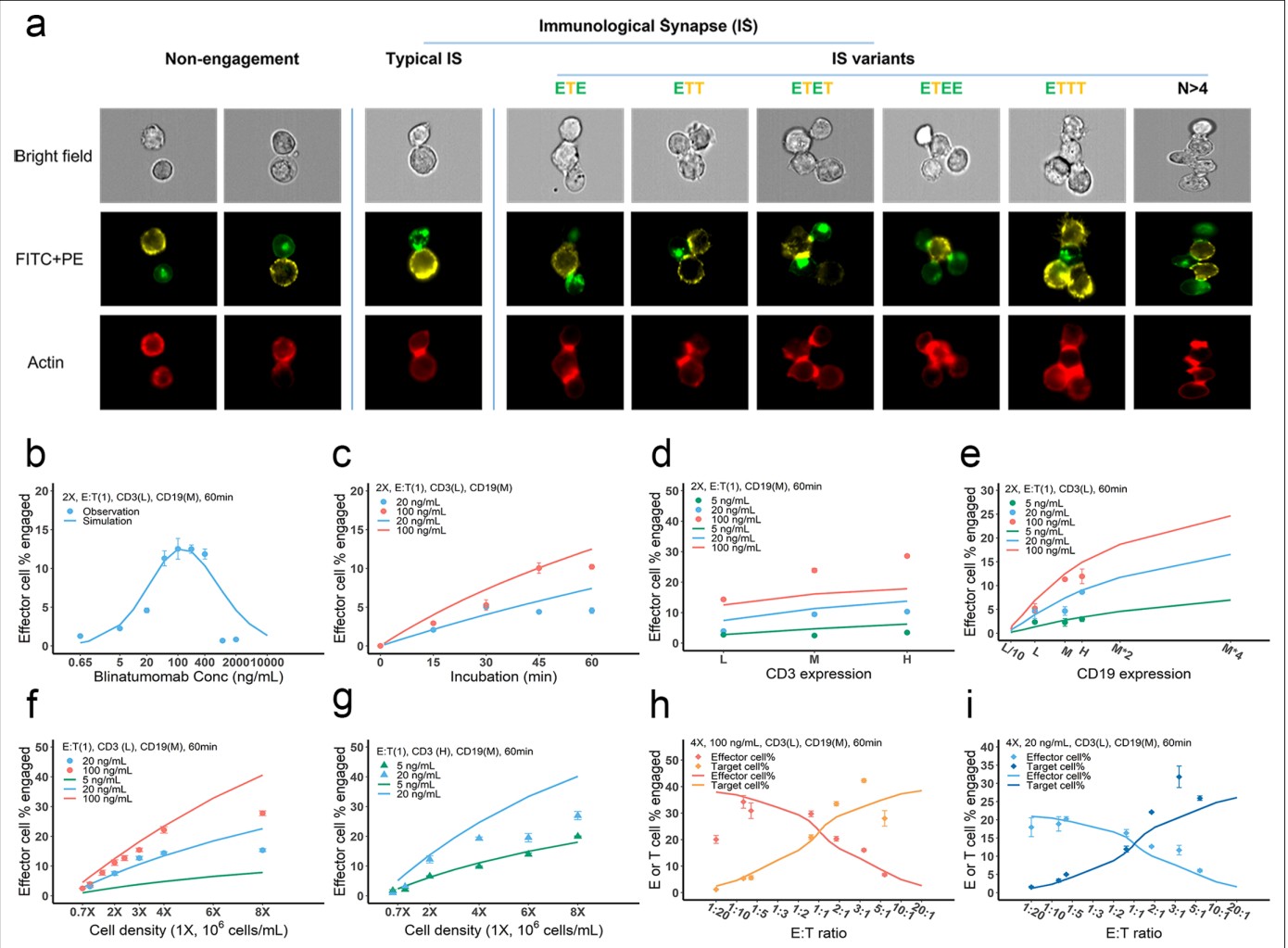

**Figure 3.** Dynamics of immunological synapse (IS) formation induced by bispecific T cell engager (BiTE) under different conditions. (**a**) Representative image of non-engagement (futile encounter), typical IS, and other IS variants. Green (FITC), effector cells (E); Yellow (PE), target cells (T). (**b–i**), The effects of drug concentration (**b**), incubation duration (**c**), antigen density (**d, e**), cell density (**f, g**), and E:T ratio (**h, i**) on IS formation. The base model was applied to simulate IS formation under different conditions. Observations are dots (with SE) and model simulations are solid curves. 2 X, 2 × 10⁶ total cells/mL; E:T(1), E:T ratio = 1; CD3(L), CD3 expression (Low); CD3(H), CD3 expression (High); CD19(M), CD19 expression (medium); 5, 20, 100 ng/mL, blinatumomab concentration; 60 min, incubation duration. All samples were biologically triplicates.

The online version of this article includes the following figure supplement(s) for figure 3:

**Figure supplement 1.** Performance of the base model (observation vs simulation).

**Figure supplement 2.** The effect of binding affinity on bispecific T cell engager (BiTE)-mediated cell-cell engagement.

## Effects of BiTE concentration, cell density, and antigen expression on IS formation

IS formation dynamics were quantified by imaging flow cytometry. Representative images of non-engaged (futile contact), typical IS, and multiple IS variants are shown in *Figure 3a*. Contact between the effector and target cells was evaluated in brightfield and FITC + PE channels. Their interfaces were classified as bona fide IS when there was a high intensity of actin (red) at the contact site, as F-actin is known to polymerize and locally concentrate at sites of the interface (*Dustin and Cooper, 2000*).

To investigate the key influential factors of IS formation, we explored multiple experimental conditions by varying BiTE concentration (0.65–2000 ng/ml), incubation duration (0–60 min), antigen expression (three levels for either CD3 or CD19), cell density (0.7–8 million total cells/mL), and E:T ratio (0.05–6) (*Figure 3b–i*). The fraction (%) of effector cells engaged in IS was quantified to inform IS

formation dynamics. We also ran these experiments virtually using the base model to test the model's predictive performance and to explore mechanistic hypotheses.

In *Figure 3b–i*, the observations (symbols) and model simulations (lines) overlapped, indicating good base model performance. IS formation in vitro exhibited a bell-shaped relationship to BiTE concentration (*Figure 3b*). The model predicted this bell-shaped relationship and revealed that high BiTE concentrations (>100 ng/ml) would reduce the formation of ternary complexes, partly because individual antigens (CD3 and CD19) were almost completely occupied by one arm of the BiTE, limiting crosslinking with opposing cells (*Appendix 1—figure 2*). IS formation increased over time and plateaued around 60 min (*Figure 3c*). We, therefore, restricted our incubation to 60 min considering IS quantification could be biased by serial cell-cell engagements and potential cell lysis (*Fousek et al., 2021*).

The effect of CD3 expression on IS formation was relatively small, especially at low BiTE concentrations (*Figure 3d*). The model revealed that only a small fraction of CD3 was occupied; therefore, we concluded CD3 expression was not a key driver of IS formation at low BiTE concentrations. The base model underpredicted the effect of CD3 expression on IS formation at 100 ng/ml BiTE concentration, which is partially because of the rapid CD3 downregulation upon BiTE engagement and assay variation across experimental conditions. In contrast, we found CD19 expression on target cells profoundly impacted IS formation (*Figure 3e*). These results were also predicted by the base model.

Our model predicted that cell density would also be critical to IS formation on a per-cell basis. Increasing total cell density (E+T) from 1 to 8 million per ml at an E:T ratio around 1:1 drastically boosted IS formation from 3.1 to 15.3% at 20 ng/ml BiTE and 3.9 to 27.8% at 100 ng/ml BiTE (*Figure 3f*). IS formation was further increased with high CD3-expressing effector cells at high cell densities (*Figure 3g*).

The E:T ratio also played a pivotal role in IS formation. Changing the E:T ratio led to variations in the fraction of effector and target cells involved in IS formation, as predicted by the model. With higher E:T ratios, a greater fraction of target cells but a lower fraction of effector cells was involved in IS formation (*Figure 3h and i*).

Overall, we found multiple factors to be influential to IS formation. The model reasonably recapitulated IS formation dynamics under various conditions (*Figure 3b–i*). The goodness of model predictions is provided in *Figure 3—figure supplement 1*. With good model predictability, we further investigated the influence of CD3 and CD19 binding affinities (*Figure 3—figure supplement 2a and b*). Counterintuitively, higher affinities to CD3 resulted in higher predicted IS formation at low BiTE concentration (e.g. 0.58% and 0.41% at 0.65 ng/mL with $K_{D,CD3}$ = $2.6×10^{-10}$ and $2.6×10^{-7}$ respectively, *Figure 3—figure supplement 2b*), but lower predicted IS formation at high BiTE concentration, which is perhaps due to the oversaturation of both CD3-BiTE and CD19-BiTE and higher induction of CD3 downregulation. Reduced IS formation at high CD3 affinities also resulted in a bell-shaped relationship (*Figure 3—figure supplement 2b*). Notably, there are papers reporting that high CD3 affinity may result in negative effect on BiTE safety and clinical efficacy (*Chen et al., 2021*; *Dang et al., 2021*). Our model suggested that blinatumomab has an affinity for CD3 within the optimal range of $10^{-7}$ – $10^{-6}$ M. In contrast, BiTEs with higher affinity to CD19 were predicted to enhance IS formation (*Figure 3—figure supplement 2c and d*).

## IS variants were prevalent and well-predicted by the base model

Many types of IS variants were observed in the experimental system. In total, six types of IS were quantified, including typical IS (ET), ETE, ETT, ETET, ETEE, and ETTT. IS variants with more than four cells were not analyzed in our study, nor included in the base model, due to their low abundance. The frequency of these variants was recorded and compared under each experimental condition.

Depending on the experimental condition, approximately 12–25% of IS observed were IS variants, although this increased up to 50% under condition 5 due to high cell density and BiTE concentration (*Figure 4a*). Among these IS variants, ETE and ETT were the most frequently observed, accounting for more than 60% of total IS variants formed under all conditions (*Figure 4b*). Conditions 5 and 11 showed high ETE frequencies due to a high E:T ratio (6:1), while conditions 6 and 12 showed high ETT frequencies due to a low E:T ratio (1:5.8). The base model well predicted the relative fraction of each IS variant under all tested conditions (*Figure 4a, b*). In general, the fraction of IS variants increased

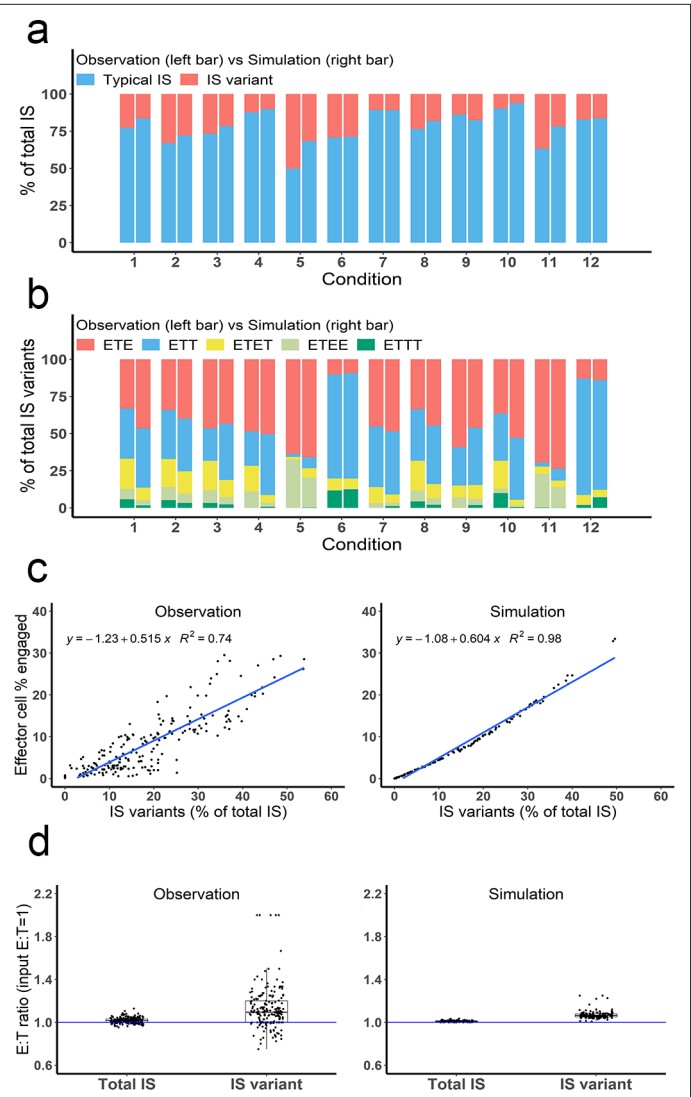

**Figure 4.** Multiple types of immunological synapses (IS) variants were observed and well-predicted by the base model. In total, six types of IS were quantified, including typical IS (ET), ETE, ETT, ETET, ETEE, and ETTT. (**a**) The fraction of typical IS and variants under different conditions; (**b**) The composition of IS variants (ETE, ETT, ETET, ETEE, ETTT) under different conditions; (**c**) The positive correlation between the fraction of IS variants (% of total IS) and total IS formation (effector cell % engaged). The formula and R² of linear regressions are shown. (**d**) The E:T ratios involved in total IS and IS variants. Experimental setup: Condition 1, 2 X, E:T(1), CD3(L), CD19(M), 100 ng/mL, 60 min; Condition 2, 4 X, E:T(1), CD3(L), CD19(M), 100 ng/mL, 60 min; Condition 3, 2 X, E:T(1), CD3(H), CD19(M), 100 ng/mL, 60 min; Condition 4, 2 X, E:T(1), CD3(L), CD19(L), 100 ng/mL, 60 min; Condition 5, 4 X, E:T(6), CD3(L), CD19(M), 100 ng/mL, 60 min; Condition 6, 4 X, E:T(0.17), CD3(L), CD19(M), 100 ng/mL, 60 min; Conditions 7–12 are the same as Conditions 1–6, except with lower bispecific T cell engager (BiTE) concentrations (20 ng/mL).

with total IS abundance. The positive correlation between the fraction of IS variants and effector cells involved in IS was well predicted by the base model (**Figure 4c**).

The E:T ratios of IS variants from all co-incubation samples were pooled for comparison (**Figure 4d**). The median E:T ratio in total IS was about 1.0. When excluding typical IS, this ratio increased to 1.1 for the remaining IS variants, suggesting slightly more effector cells were involved in IS variant formation than target cells, in line with model predictions.

## The in vitro model predicted antigen escape and organ reservoirs

Effector T cells detach from IS and re-engage with other target cells in a process called serial cellular engagement. These effector T cells are also known as 'serial killers' (*Fousek et al., 2021*; *Rogala*

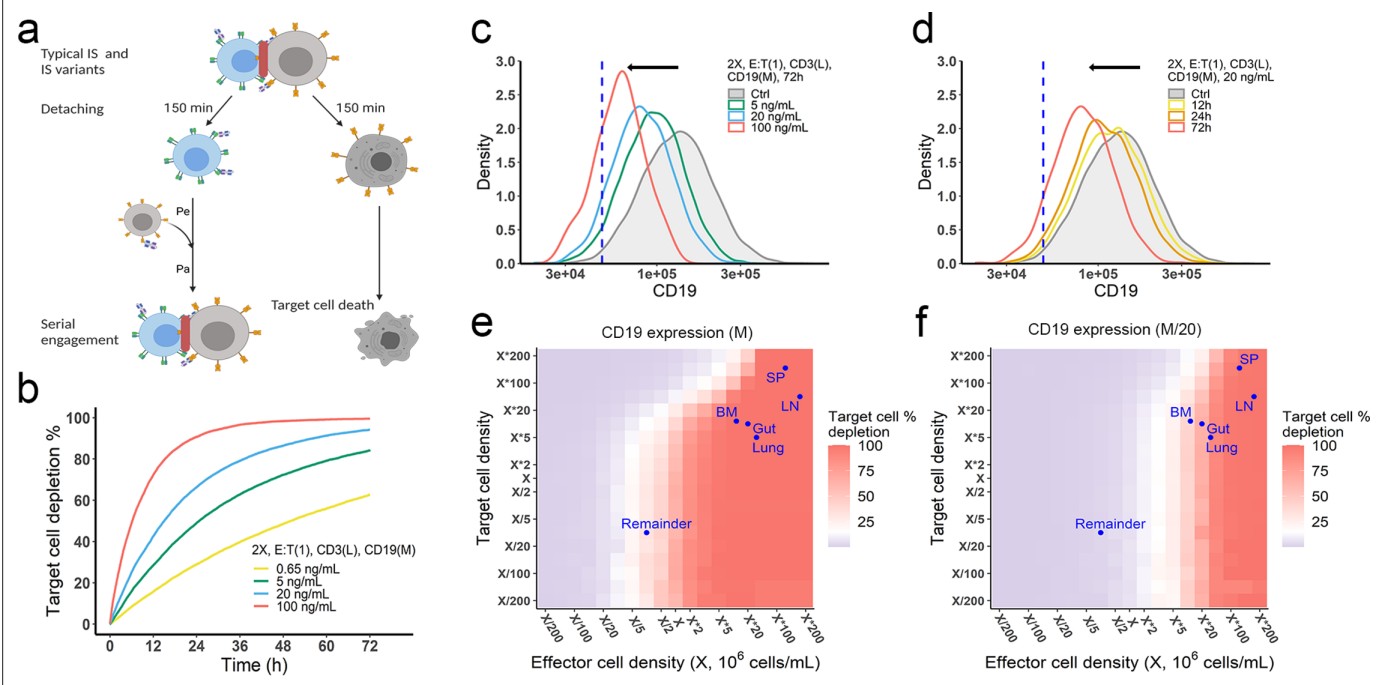

**Figure 5.** The in vitro model predicted tumor evolution in time and space. (**a**) Scheme of cell detachment and serial engagement in the in vitro model. Immunological synapses (IS) duration is set to 150 min. Pe, encounter probability, Pa, adhesion probability; (**b**) Long-term simulation (72 hr) of target cell depletion across drug concentrations; (**c–d**) The effects of drug concentration (**c**) and incubation time (**d**) on CD19 expression. Dashed line, a pre-defined threshold value of CD19 expression for 15% target cell depletion within 72 hr (initial setup: 2 X, E:T(1), CD3(L), 0.65 ng/mL, 72 h). Ctrl, the initial distribution of CD19 expression in the target cell population. (**e–f**) the effects of effector and target cell density on target cell depletion (%). Dots indicated the effect and target cell densities in healthy human organs. White color, 15% target cell depletion. BM, bone marrow; LN, lymph nodes; SP, spleen; Remainder, all the rest of the non-lymphoid organs. Initial setup: CD3(L), CD19(M) for (**e**), CD19 (M/20) for (**f**), 0.65 ng/mL, 72 h.

The online version of this article includes the following figure supplement(s) for figure 5:

**Figure supplement 1.** The effects of ET ratio (**a**), cell density (**b**), and binding affinity (**c**) on CD19 evolution.

**Figure supplement 2.** The effects of effector and target cell density on target cell depletion (%) at 72 h.

**Figure supplement 3.** Different effects of CD19 on cellular and molecular processes.

*et al., 2015*). We extended the base model to incorporate IS detachment and re-engagement (*Figure 5a*). The in vitro model simulated IS formation and cellular cytotoxicity for up to 72 hr. With serial engagement and killing, the fraction of target cell lysis increased considerably, even at low BiTE concentrations (*Figure 5b*).

Importantly, the in vitro model predicted tumor evolution toward populations with low CD19 expression (i.e. antigen escape). Approximately 10–20% of patients who relapse after blinatumomab treatment experience antigen escape, which decreases the efficacy of subsequent anti-CD19 CAR-T cell therapy (*Braig et al., 2017*; *Pillai et al., 2019*). As shown in the model, tumor cells with lower CD19 expression had a lower chance of being engaged by effector cells and thus a higher probability of surviving (*Figure 5c, d*). The speed of evolution was predicted to increase at greater BiTE concentrations (*Figure 5c*) and accelerate over time (*Figure 5d*). The effect of E:T ratio, cell density, and antigen affinity on tumor evolution were also simulated (*Figure 5—figure supplement 1*). Notably, greater IS formation led to more extensive evolution toward lower CD19-expressing cells.

The impact of cell density at clinically relevant BiTE concentrations was also interrogated (*Figure 5e, f, Figure 5—figure supplement 2*). Notably, an increase of effector cell density resulted in higher fractions of target cell lysis at 72 hours. However, a higher density of target cells did not markedly diminish the fraction of target cells lysed at a given effector cell density, due to a compensatory increase in the probability of cell-cell encounter probability per effector cell. When target cell density was extremely high (e.g., > 5×10⁶/mL with medium CD19 expression at 1.45×10⁵/cell in *Figure 5e*), lysis fraction decreased, as the low BiTE concentration may have become a limiting factor for IS formation. We used

organ-specific effector and target cell abundance (*Supplementary file 1b*) to compare the predicted gradient of cell lysis across organs (*Figure 5e, f*, *Figure 5—figure supplement 2*). Higher target cell lysis was predicted in lymph nodes and the spleen due to the abundance of effector cells in these organs. The bone marrow and all the rest of non-lymphoid organs (the remainder) showed restricted cell lysis primarily due to their relatively low abundance of effector cells (*Figure 5f*). The model also predicted that some organs like the bone marrow may become tumor cell sanctuary sites, providing space for tumor cell survival and adaptation, thereby increasing the likelihood of treatment resistance.

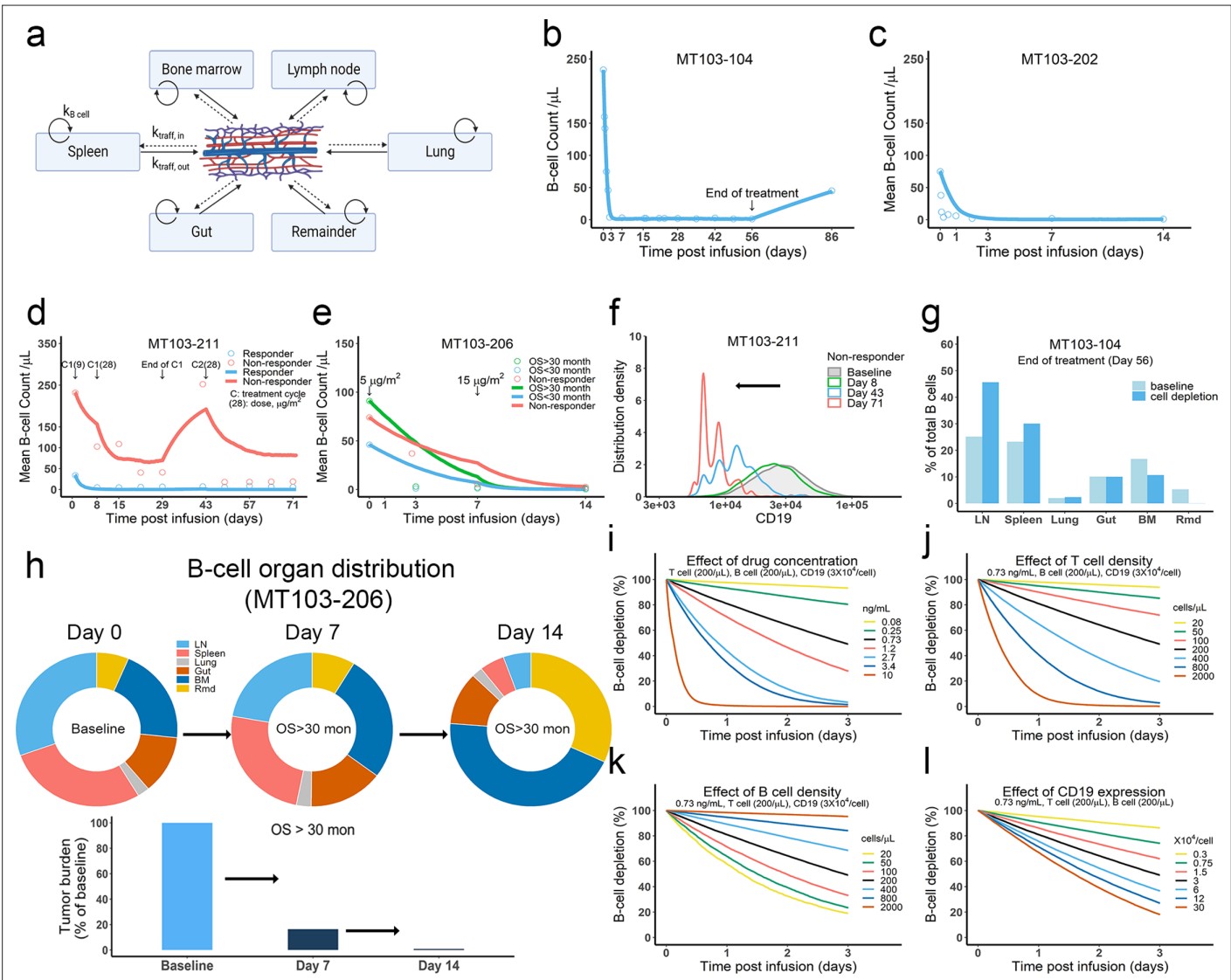

**Figure 6.** The in vivo model predicted clinical pharmacodynamics and tumor evolution across anatomical sites. (**a**) Scheme of organ compartment and cell trafficking. Remainder, all the rest of the non-lymphoid organs; $k_{B\ cell}$, the turnover rate of B cell; $k_{traff,\ in}$ and $k_{traff\ out,}$ B cell trafficking rate. For parameters and trial information, see Materials and methods, and *Supplementary file 1b and c*; (**b–e**) Observed and simulated patient B cell profiles in blood; (**f**) Simulated CD19 evolution in non-responder patients of trial MT103-211; (**g**) Simulated cell lysis potency for each organ in trial MT103-104; (**h**) Simulated baseline (day 0), and post-treatment (day 7 and 14) B cell organ distribution in patients with OS >30 months of trial MT103-206. Bar plot, simulated baseline and post-treatment tumor burden; (**i–l**) Sensitivity analyses for the impact of drug concentration (**i**), T cell density (**j**), B cell density (**k**), and CD19 expression (**l**) on B cell depletion. T cell density change is allowed in the simulations (**b–e**), details see *Supplementary file 1b*. BM, bone marrow; LN, lymph nodes; OS, overall survival; Rmd, remainder.

The online version of this article includes the following figure supplement(s) for figure 6:

**Figure supplement 1.** The effect of blood flow on the production and stability of immunological synapses (IS).

**Figure supplement 2.** Simulated baseline (day 0), and post-treatment (day 7 and 14) B cell organ distribution in patients of trial MT103-206.

## The in vivo model predicted clinical outcomes and tumor evolution across anatomical sites

We developed the in vivo model by defining IS formation dynamics in organs and cell trafficking across organ compartments (*Figure 6a*, Materials and methods, *Supplementary file 1b and c*). We used the model to simulate cell lysis in each organ and tumor-killing profiles throughout the body. Organ-specific cell lysis is highly dependent on relative IS formation dynamics and thus is a function of organ-specific effector (T cell) and target (B cell) populations, as well as BiTE exposure.

In the in vivo model, the blood compartment serves merely as a trafficking route and does not mediate IS formation and detachment (*Figure 6a*). This assumption was supported by our observation that negligible IS was formed under shear stress forces approximating those experienced under blood flow (*Figure 6—figure supplement 1a*). Once formed, IS in the blood are unlikely to be broken through shear stress (*Figure 6—figure supplement 1b*). Blood B cell levels reflected the systemic average. Although only 2% of lymphocytes are present in the blood, blood flow can transport about 5 $\times 10^{11}$ lymphocytes each day – comparable to the total number of lymphocytes in the body (*Westermann and Pabst, 1992*).

Patients show mixed responses to BiTE immunotherapy. Some patients exhibit complete tumor eradication while others have negligible responses. By adopting patient-specific parameters, such as BiTE dosing regimens and T cell proliferation profiles, the in-vivo model reasonably predicted B-cell depletion profiles in patients (*Bargou et al., 2008*; *Klinger et al., 2012*; *Zhu et al., 2016*; *Zhu et al., 2018*; *Zugmaier et al., 2015*) treated with blinatumomab in multiple clinical trials (*Figure 6b–e*, *Supplementary file 1b,c*). In the trials MT103-211 and MT103-206, rapid accumulation of T cells in the blood was observed in responders but not non-responders (*Zhu et al., 2018*; *Zugmaier et al., 2015*). The model could account for these patient-specific T cell profiles and distinguish between responding and non-responding patients (*Figure 6d, e*).

Like the in vitro model, the in vivo model also predicted evolution toward low CD19-expressing cell populations over time, as shown in non-responders in MT103-211 (*Figure 6f*). This process is inevitable; the stronger the therapeutic pressure, the lower CD19-expression in the surviving cell population. The fraction of low CD19-expressing cells increases over time while the efficiency of tumor cell lysis decreases, leading to a gradual loss of drug sensitivity.

We finally explored tumor evolution across anatomical sites and characterized the spatial gradients of cell lysis. The lymph node, spleen, and lung showed higher fractions of cell lysis than the gut, bone marrow, and remainder (*Figure 6g*). More than 45% of malignant cells in the system were lysed in the lymph nodes and around 30% were eradicated in the spleen. Lytic fractions were higher than their respective baseline levels in both organs, confirming enhanced tumor killing mediated by BiTE. In contrast, the lytic fraction in the bone marrow was lower under treatment than at baseline (*Figure 6g*), indicating poor tumor lysis efficiency. The anatomical differences in the efficiency of cell lysis affected B cell biodistribution after BiTE treatment in patients (*Figure 6h*, *Figure 6—figure supplement 2*). The relative anatomical distribution of B cells also shifted considerably over time. In high responders (OS >30 months, MT103-206), over 99% of B cells were eradicated, particularly in organs with high predicted lysis (lymph nodes and spleen). In the bone marrow, a small fraction of B cells survived that exhibited considerably lower CD19 expression than the original cell population. Unfortunately, the surviving cell populations gradually repopulated the bone marrow, leading to B cell rebound and eventually patient relapse. By contrast, non-responders had a lower fraction of B cell lysis by day 14, with B cell distribution profiles remaining similar to the baseline (*Figure 6—figure supplement 2*).

Sensitivity analyses confirmed that baseline tumor burden, drug concentration, cytotoxic T cell infiltration, and CD19 expression were critical to patient response (*Figure 6i - l*).

## The in vivo model predicted optimal dosing regimens for blinatumomab

We applied the in vivo model to simulate B cell-killing efficacy and CD19 evolution during blinatumomab treatment and compared different doses and regimens (*Figure 7a*). The initial plasma B cell abundance was assumed to be 200 cells/μL, with varying levels of growth rates. Under the approved dose (i.e. the high dose) and scheme 1, tumor-killing profiles were highly dependent upon tumor growth rate and baseline T cell abundance (*Figure 7b*). Tumors gradually accumulated resistance to treatment, especially fast-growing tumors. For slow-growing tumors with low T cell baseline, the medium dose showed a comparable tumor-killing effect but resulted in less CD19 evolution than the

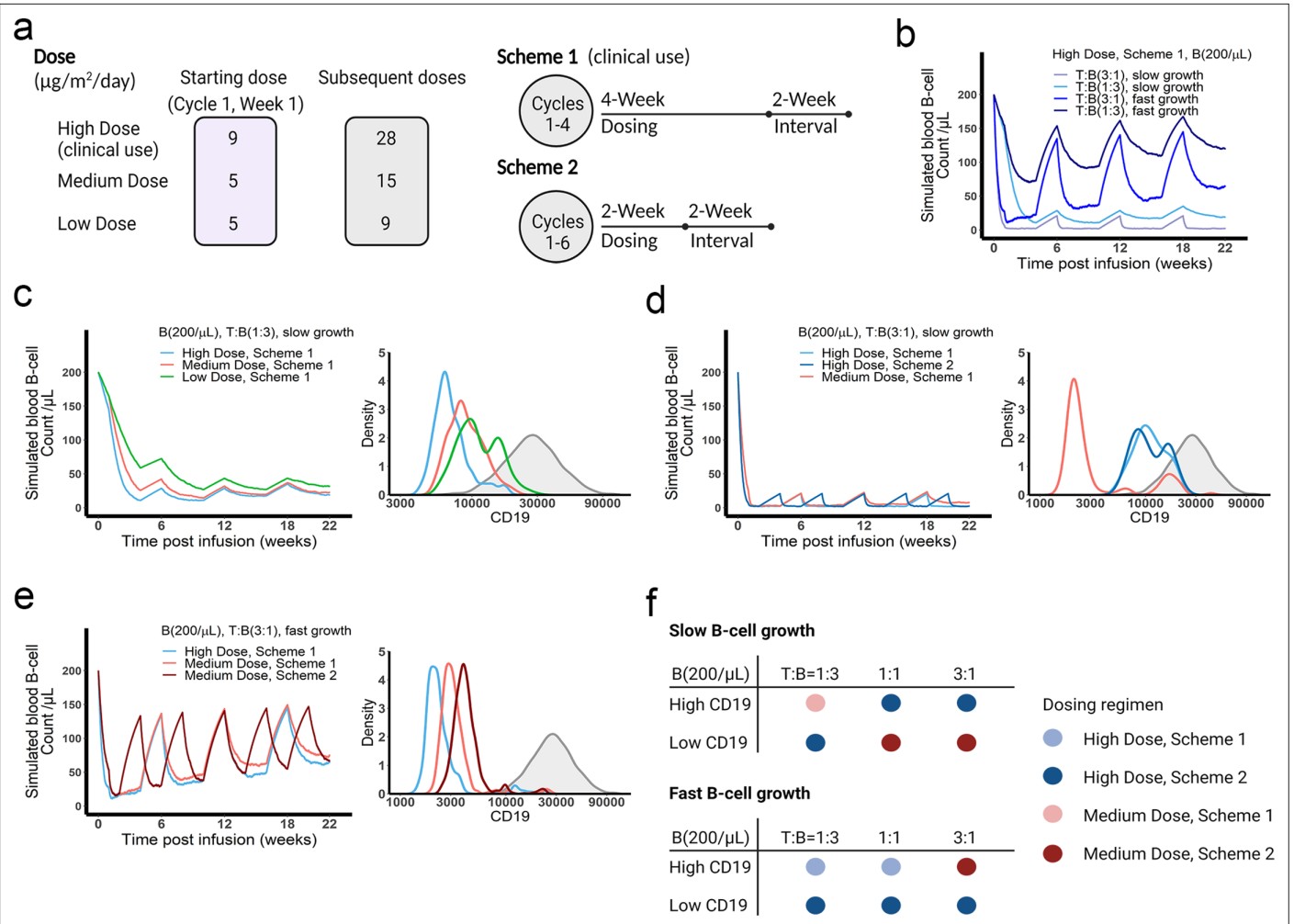

**Figure 7.** The dose and regimen of bispecific T cell engager (BiTE) strongly influenced tumor control and CD19 evolution. (**a**) Three doses (high, medium, and low) and regimens (scheme 1 and 2) were evaluated in the simulations. Starting doses were applied in the first week of cycle 1 only. The high dose with scheme 1 is the clinically approved dose and regimen of blinatumomab for the treatment of B cell acute lymphoblastic leukemia; (**b**) Simulated blood B cell profiles under different T:B ratios and B cell growth rates in 22-week treatment; (**c–e**) Different dose levels, and schemes were explored in respective conditions. Simulated blood B cell profiles and CD19 evolution were shown. Gray line and shaded regions represent baseline CD19 expression; (**f**) The favorable dosing regimen under each condition. The favorable dosing regimen was determined by comparing B cell killing efficacy, CD19 evolution, and total dose, in this order of priority, respectively. B (200 /μL), baseline B cell density in blood is assumed to be 200 /μL; High (**b–e**) and low CD19 expressions were the mean level of CD19 expression per B cell and set as $3 \times 10^4$ and $1 \times 10^4$, respectively; High growth and low growth of B cells were set to 0.071 /day and 0.0071 /day; T cell density change is not included in this proof-of-concept simulations.

high dose (*Figure 7c*). In contrast, for slow-growing tumors with high T cell abundance, the high dose exhibited almost complete tumor control and much less CD19 loss than the medium dose (*Figure 7d*). The high dose showed similar efficacy at the two dosing schemes, but scheme 2 had fewer total doses. For fast-growing tumors, CD19 loss was significant, regardless of dose and regimen (*Figure 7e*). The medium dose in scheme 2 elicited less CD19 loss and better tumor control than scheme 1. *Figure 7f* summarized the favorable dosing regimen under each condition. We found that the approved dose or regimen was suboptimal for most slow-growing tumors; rather, the medium dose or dosing scheme 2 could reach similar efficacy with slower CD19 evolution. The high dose was required for almost all fast-growing tumors, with the only exception being patients with high CD19 expression and high T cell abundance at baseline who received more benefit from the medium dose at scheme 2. Overall, our in vivo model, through defining IS formation dynamics across anatomical sites in the system, could predict BiTE pharmacodynamics and changes in CD19 expression over time, and identify optimal dosing strategies based on baseline tumor characteristics.

## Discussion

The clinical efficacy of BiTE immunotherapy remains suboptimal, with many of the patients who initially respond eventually experiencing disease relapse. Understanding IS formation, a crucial step in many T cell immunotherapy's mechanisms of action, can yield insights into the pharmacodynamics of BiTEs and subsequent treatment resistance. This study used an experimentally and theoretically integrated approach to examine IS formation dynamics induced by BiTEs on a population level. The abundance of IS caused by BiTEs was quantified using imaging flow cytometry, and the dynamics of IS formation were simulated with theoretical models. By defining IS formation as a spatiotemporal orchestration of molecular and cellular interactions, our theoretical models recapitulated the experimental data well. Notably, the models predicted antigen escape to be a common mechanism of resistance to BiTE immunotherapy. Tumor cells with low antigen expression accumulated over time, leading to treatment resistance and eventual disease relapse. The anatomical heterogeneity of T cell infiltration and E:T ratios across organs also conferred heterogeneous degrees of cell lysis. In particular, a subset of tumor cells in 'sanctuary sites,' such as the bone marrow, may be relatively protected from effector cell lysis and fuel tumor evolution and disease relapse.

IS formation induced by BiTEs is determined mainly by two cell-scale interaction processes: cell-cell encounter and adhesion. Cell-cell encounter is the first, and in many instances, the rate-limiting step to IS formation. For simplicity, the models assume random cell motion without consideration for directed or chemotactic movement (*Celli et al., 2012*). Cell density is another critical factor; cell encounter probability could become the rate-limiting step for IS formation when the target cell density is sufficiently low that effector cells have little chance of encountering target cells. This could be particularly challenging for patients with minimal residual disease. When effector cell density is low, as in the bone marrow, tumor cells have a higher chance of surviving for long enough to develop immune evasion mechanisms, leading to treatment resistance. On the other hand, when target cell density is extremely high within an organ, target cell lysis may be compromised by insufficient antibody concentrations at clinically utilized doses (*Figure 5e and f*). This is consistent with clinical observations that the efficacy of blinatumomab is much higher in patients with relatively low tumor burden (*Topp et al., 2015*; *Viardot and Bargou, 2018*).

Synapse formation is a set of precisely orchestrated molecular and cellular interactions. Our model merely investigated the components relevant to BiTE pharmacologic action and thereby serve as a simplified representation of this process. The molecular crosslinking between BiTEs and antigens affects the probability of cell-cell adhesion upon encounter. Drug concentration, binding affinity, and antigen expression are critical determinants of this process. Adhesion molecules such as CD2-CD58, integrins, and selectins, are critical for cell-cell interaction. The model did not consider specific roles played by these adhesion molecules, which were assumed constant across cell populations. The model performed well under this simplifying assumption.

Many studies have reported a bell-shaped drug concentration-response profile for BiTE immunotherapy (*Betts et al., 2019*; *Douglass et al., 2013*; *Schropp et al., 2019*; *Van De Vyver et al., 2021*), with the primary mechanism underlying the phenomenon being the oversaturation of T cell receptors at high BiTE concentrations. The theoretical model reported herein also predicts a bell-shape concentration – IS curve, but the predicted curve peaked at higher antibody concentrations if not including the possibility of CD3 down-regulation by effector cells upon antibody engagement.

We explored the different effects of CD19 on cellular and molecular processes. Total CD19 in the system was jointly influenced by CD19 expression on membrane and target cell density. CD19 expression influenced cell lysis to a similar extent as target cell density when both factors were low (*Figure 5—figure supplement 3a*). However, an increase of CD19 expression beyond $3 \times 10^5$ receptors/cell did not further improve cell lysis. In contrast, target cell densities seemed to have a bidirectional effect on cell lysis. At low levels, escalating cell densities enhanced the probability of cell-cell encounter, while at high target cell densities, BiTE concentrations were insufficient to mediate meaningful IS formation, resulting in fewer cell-cell adhesion events and less cell lysis (*Figure 5—figure supplement 3a, b*). This is consistent with clinical simulations (*Figure 5—figure supplement 3c*). The different roles of CD19 expression and target cell density highlight the importance of cellular-scale interactions to IS formation that cannot be appropriately described by molecular crosslinking alone. Because of these cellular processes, our theoretical models fundamentally differ from previous BiTE pharmacodynamics models that consider molecular crosslinking only (*Betts et al., 2019*; *Jiang et al.,*

*2018*; *Schropp et al., 2019*; *Song et al., 2021*). In our models, molecular crosslinking caused by BiTE, i.e., ternary complex formation, drove cell-cell adhesion events, whereas cell-cell encounters were modeled as an independent process.

Heterogeneity of CD19 antigen expression is a critical factor in BiTE-induced IS formation. Target cells with lower antigen expression had a lower probability of adhesion to T cells and thus a greater chance of survival. The theoretical models suggest that tumor evolution is an inevitable consequence in treatment, and that the stronger the therapeutic selection pressure, the more tumor cell populations evolve away from their pretreatment phenotype. Ultimately, the surviving tumor cells shift toward a low antigen expression population in a process known as antigen escape. Antigen escape is a common mechanism of resistance to T cell-based immunotherapy (*Aldoss et al., 2017*; *Mejstríková et al., 2017*; *Samur et al., 2021*; *Topp et al., 2014*; *Xu et al., 2019*); however, the speed of tumor evolution toward antigen escape remains hard to predict. Through defining the formation of IS, our models show a proof of concept for predicting the trajectory of antigen escape based on baseline antigen expression, more validations of our models are warranted.

Non-uniform tumor lysis effect across organs represents another barrier for therapy. Provided an anatomical space with few effector cells, tumor cells might use the bone marrow as a sanctuary site within which IS formation is infrequent. Insufficient selective pressure from effector cells might allow the regeneration of a newly resistant population of tumor cells that then repopulate other organs and accelerate systemic disease progression. This speculation is consistent with the clinical observation that patients under BiTE treatment often have relapses first detected in the bone marrow (*Locatelli et al., 2022*). Of note, the inadequate tumor lysis in the bone marrow might also be explained by tissue-specific differences in chemokine gradients that hinder cell-cell interaction and adhesion.

We used the in vivo model to compare different doses and schedules of blinatumomab. We found that tumor baseline characteristics, including tumor growth rate, CD19 expression, and T cell abundance, greatly influenced tumor-killing pharmacodynamics, tumor evolution, and consequentially, the ideal dosing regimen. The clinically approved dose and regimen might become suboptimal for most slow-growing tumors. The medium dose or mild regimen could maintain an optimal balance between tumor-killing and evolutionary pressure (*Figure 7f*). However, it was not always the case that higher dose amounts (i.e. higher therapeutic pressure) resulted in faster tumor evolution. For slow-growing tumors with sufficient T cells (*Figure 7d and f*), the high dose with regimens 1 or 2 could cause nearly complete tumor eradication, thereby resulting in negligible selection and limiting the total population size remaining for evolution. Faster and greater reductions in population size conferred by high doses might, therefore, reduce the chance of evolutionary rescue for slow-growing tumors.

There are still some limitations to our models. Specifically, our model operates under the assumption that target cells will be eradicated upon the formation of IS. However, further improvements to our model that consider the varying killing efficiencies of different IS variants may enhance its clinical relevance and overall performance. For IS formation and T cell motility pattern: our models only considered a few select factors that influence the formation of IS, which may not provide a full description of drug inhomogeneous efficacy across anatomical sites. Factors like the heterogeneous distribution of T and B cells, chemokines, and stromal structures could affect the T cell motility and functions in tissue environments, and including these factors may provide an unbiased evaluation of drug effect across tissues. We assumed Brownian motion in the model as a good first approximation of T cell movement. However, T cells often take other more physiologically relevant searching strategies closely associated with many stromal factors. Because of these stromal factors, the cell-cell encounter probabilities would differ across anatomical sites. For T cell activation: our models did not include intracellular signaling processes, which are critical for T activation and cytotoxicity. However, our data suggest that encounter and adhesion are more relevant to initial IS formation. To make more clinically relevant predictions, the models should consider these intracellular signaling events that drive T cell activation and cytotoxic effects. Of note, we did consider the T cell expansion dynamics in organs as an independent variable during treatment for the simulations in *Figure 6*. T cell expansion in our model is case-specific and time-varying. For model parameters: the majority of model parameters were obtained or derived from the literature, and we did not perform model optimization to get the optimal values of model parameters. The only parameter we manually optimized is the sensitive coefficient for cell-cell adhesion in the base and in vivo model and the values were calibrated against

the in vitro data. Implementing model optimization algorithms would improve the predictability of the models.

In conclusion, our study investigated the dynamics of IS formation under various conditions mimicking the heterogeneous nature of tumor microenvironments. To our knowledge, these theoretical models are the first to quantify the entire BiTE-induced IS formation process. The models reveal trajectories of tumor evolution through antigen escape across anatomical sites and suggest dosing regimens that could confer tumor control in light of treatment-induced disease evolution. This work has substantial implications for T cell-based immunotherapies.

## Materials and methods

### Cell lines

Jurkat (Clone E6-1) and Raji cells were obtained from ATCC and maintained in RPMI1640 supplemented with 10–20% fetal bovine serum (FBS) and 1% penicillin-streptomycin. Cell lines were routinely tested to avoid mycoplasma contamination. Cell lines have been authenticated by STR profiling and no mycoplasma contamination was detected.

### Cell sorting and antigen expression quantification

Cell populations with high (H), medium (M), and low (L) antigen expression were sorted based on natural expression levels, without any genetic engineering. PE-anti-CD3 and PE-anti-CD19 (BD Biosciences, San Jose, CA) were used as staining antibodies and BD FACSAria II was used to perform cell sorting. Surface expression of CD3 and CD19 was quantitatively determined by Quantum MESF beads (Bangs laboratories, Fishers, IN) and BD LSR II flow cytometry (*Figure 1—figure supplement 1*).

### Cell co-incubation and imaging flow cytometry

Effector cell (E, Jurkat), target cell (T, Raji), and anti-hCD19-CD3 BiTE (BioVision, Milpitas, CA) were well mixed and co-incubated in 1 mL medium at 37 °C. CD3 or CD19 expression, drug concentration, cell density, E:T ratio, and duration of co-incubation varied as initial setups. All samples were biological triplicates. After co-incubation, the effector cell, target cell, actin, and nucleus were stained by FITC-anti-CD7 (eBiosciences, San Diego, CA), PE-anti-CD20 (BD Biosciences, San Jose, CA), AF647-anti-phalloidin (Thermo Fisher, Waltham, MA) and DAPI, respectively. Staining for surface and intracellular markers was performed as described previously (*Liu et al., 2019*). Samples were analyzed using Amnis ImageStream MKII (Luminex, Austin, TX). The frequency of IS was quantified using IDEAS (Luminex). The gating strategy is summarized in *Figure 1—figure supplement 2*.

### Cell co-incubation under shear stress

To mimic the shear stress in blood circulation, the effector cell, target cell, and BiTE were well mixed in a circular pipe (internal diameter: 1.6 mm) and co-cultured (37 °C) at a certain flow velocity driven by a roller pump (Masterflex, Vernon Hills, IL). Flow velocities were adjusted to produce varying wall shear stresses, equivalent to those in the vein (1–6 dyn/cm$^2$), artery (10–24, dyn/cm$^2$), and capillary (20–40 dyn/cm$^2$) (*Kamiya et al., 1984*; *Papaioannou and Stefanadis, 2005*; *Sebastian and Dittrich, 2018*). After co-culture, sample staining and analysis were the same as previously described.

### Modeling and simulation

#### Base model

Mechanistic agent-based models were developed to simulate IS formation dynamics in vitro and in vivo. We employed a sequential model-building strategy. First, we developed a model to capture IS formation dynamics within 1 hr, called the base model. The base model consisted of three modules at different dimensions: antibody-antigen binding (3D), cell-cell encounter, and cell-cell adhesion (2D) (*Figures 1 and 2*).

Antibody-antigen binding to form binary complexes (BiTE-CD3 and BiTE-CD19) was considered a 3D process among free molecules. Rapid equilibrium was assumed. At specific total concentrations of CD3 (A), CD19 (B), and BiTE (Y) in the co-incubation system, the equilibrium concentrations of free antigens ([A] and [B]) and binary complex ([AY] and [YB]) (Appendix 1.1) were solved according to a model with two competing binding ligands (*Yan et al., 2012*). Heterogeneous cell populations at

equilibrium were generated by randomly assigning antigens and binary complexes to each individual cell (assign A and AY to the effector cell, B and YB to target cell). The distribution of assigned antigens on the cells were consistent with measured CD3 and CD19 expressions in respective cell lines (*Figure 1—figure supplement 1*, *Supplementary file 1a*).

CD3 down-regulation on effector cells was also considered in our model (*Lanzavecchia et al., 1999*; *San José et al., 2000*; *Sousa and Carneiro, 2000*; *Utzny et al., 2006*; *Valitutti et al., 1995*; *Viola et al., 1997*). The rate of CD3 down-regulation was modeled as a function of surface binary complex abundance ([AY]'), which was described by an empirical equation that reached a steady state around 1 hr incubation, in line with the literature (Appendix 1.2). CD19 internalization was also introduced at a rate constant of 0.002 /min (*Du et al., 2008*; *Ingle et al., 2008*). Changes in surface antigen abundance ([A]', [AY]', and [YB]') for each cell were updated every 60 s in the model.

Encounters between effector and target cells are an essential step for IS formation. We adopted an approximate equation to determine the encounter probability of one effector cell meeting at least one target cell within a specific time, which is a function of the number of target cells in the system (*Celli et al., 2012*). We assumed that effector cells diffuse independently, moving in Brownian motion, while target cells were immotile. For simplicity, the IS and IS variants were considered as a singular moving cell entity when calculating the probability of encountering an additional free cell. Notably, spatial factors were considered in this encounter probability. The equations and parameters are provided in Appendix 1.3; the spatial coefficients for different encounter scenarios are listed in *Supplementary file 1a*. Encounter probabilities were re-calculated every 60 s in the model due to the changing number of free cells over time.

When an effector cell physically contacts a target cell, it may have a chance to adhere or diffuse away. Cell-cell adhesion is mediated by ternary complexes (AYB, bond) formed from binary complexes (AY or YB) and the availability of free antigens (B or A) on opposing cell surfaces (2D binding) (*Appendix 1—figure 2*). We assumed that stable adhesion relied on generating sufficient AYB bonds within a short contact duration. Differential equations were used to simulate the number of AYB bonds. The relationship between adhesion probability and the bond number was described by a modified deterministic equation (Appendxi 1.4; *Chesla et al., 1998*; *Huang et al., 2010*). A higher number of AYB bonds yielded a higher chance for engagements to result in an IS. The randomness arose from randomly assigned antigen expressions on both cells and contact duration (0.1–5 s). It is noteworthy that the 'on' and 'off' rate constants used in the equations are 2D kinetic constants on the cellular membrane, which were derived from 3D rate constants by a 'single-step model' (Appendxi 1.5; *Bell, 1978*; *Dreier et al., 2002*; *Faro et al., 2017*; *Jansson, 2010*).

In the base model (time scale ≤1 hr), each free cell had only one chance to encounter in each round (60 s). Cells that failed to encounter or adhere would remain free cells in the next round. Newly formed IS would have a chance to encounter an additional free cell to form an IS variant in the next round (*Figure 2*). Only one free cell was allowed to be added at a time. IS variants of up to four cells were allowed in the model. The algorithm for the base model is provided in *Appendix 1—figure 3*. Other important assumptions in the base model include: (1) no cell proliferation within 1 hr; (2) no change in binding equilibrium for binary complex within 1 hr; (3) once formed, IS are not breakable; (4) target cells were assumed to be eradicated after IS formation, without distinguishing between typical IS and IS variants.

## In vitro model

Next, the in vitro model was developed by incorporating serial cell engagement into the base model (*Figures 1b and 5a*). The duration of IS was assumed to be 150 min (*Fousek et al., 2021*), and the detached effector cells from IS became free cells for additional IS formation. We made the following updates and assumptions to extend the time scale to 72 hr: (1) binding equilibrium for the binary complex was recalibrated every hour; (2) CD19 internalization was ignored; (3) CD3 down-regulation remained unchanged after 1 hr; (4) BiTE concentration remained constant; (5) target cell death rate is not included as target cells were assumed to be eradicated after IS formation.

## Clinical translation (in vivo model)

Lastly, we expanded the model to develop the in vivo model for simulating tumor-killing profiles in patients. Several additional modules, including multiple organ compartments and cell trafficking across organs, were defined in the in vivo model (*Figures 1b and 6a*).

In the in vivo model, IS formation occurred within each organ-specific environment (*Supplementary file 1b*). BiTE concentrations and antigen expression (CD3 and CD19) in human T and B cells were estimated based on reported values (*Ginaldi et al., 1996*; *Ginaldi et al., 1998*; *Haso et al., 2013*; *Jiang et al., 2020*; *Ramakrishna et al., 2019*; *Rosenthal et al., 2018*). The patient-specific T and B cell densities at the organ level were derived based on the blood T and B cell numbers and the pre-defined 'partition repertoire', which reflected the relative cell abundance between blood and each organ (*Hall et al., 2012*; *Hassan and El-Sheemy, 2004*; *Westermann and Pabst, 1992*).

The rate of B cell trafficking to blood ($k_{traff, out}$) was set at 4.17% per hour, which represents the fraction of B cells trafficking through the blood during a day relative to the total B cell in the system, in line with clinical data (*Westermann and Pabst, 1992*). Each hour, 4.17% of B cells in each organ were randomly released to the blood and then reassigned to a new organ according to the pre-defined 'partition repertoire' representing trafficking through the blood. Therefore, the rate of B cell trafficking to the organ ($k_{traff, in}$) was assumed to be organ specific. T cell trafficking was not modeled, as T cell death was ignored. B cell turnover and changes in T cell densities and BiTE dose over time were allowed (*Supplementary file 1b and c*). Similar to base and in vitro model, B cell death rate was not included as B cells were assumed to be eradicated after IS formation.

As the model output, blood B cell levels after treatment was derived from the residual B cells in organs based on the pre-defined 'partition repertoire'.

## Data source, software, and code availability

Publicly available clinical data (*Bargou et al., 2008*; *Klinger et al., 2012*; *Zhu et al., 2016*; *Zhu et al., 2018*; *Zugmaier et al., 2015*) were digitized from the literature using WebPlot Digitizer. Simulation, plotting, and statistical analysis were implemented in R (3.6.0). The base model code can be found on GitHub at https://github.com/zhoujw14/BiTE-Code, (copy archived at *Zhou, 2022*).

## Acknowledgements

We thank Amgen Inc for kindly sharing blinatumomab to support our pilot study. National Institute of Health, R35GM119661.

## Additional information

### Funding

| Funder | Grant reference number | Author |
| --- | --- | --- |
| National Institute of General Medical Sciences | GM119661 | Yanguang Cao |

The funders had no role in study design, data collection and interpretation, or the decision to submit the work for publication.

### Author contributions

Can Liu, Conceptualization, Data curation, Software, Formal analysis, Validation, Investigation, Visualization, Methodology, Writing – original draft, Writing – review and editing; Jiawei Zhou, Stephan Kudlacek, Investigation, Methodology, Writing – review and editing; Timothy Qi, Software, Writing – review and editing; Tyler Dunlap, Writing – review and editing; Yanguang Cao, Conceptualization, Resources, Software, Formal analysis, Supervision, Funding acquisition, Validation, Investigation, Visualization, Methodology, Writing – original draft, Project administration, Writing – review and editing

### Author ORCIDs

Stephan Kudlacek http://orcid.org/0000-0002-0029-9984
Timothy Qi http://orcid.org/0000-0002-4359-8271
Yanguang Cao http://orcid.org/0000-0002-3974-9073

### Decision letter and Author response

Decision letter https://doi.org/10.7554/eLife.83659.sa1
Author response https://doi.org/10.7554/eLife.83659.sa2

## Additional files

### Supplementary files

• Supplementary file 1. A summary of model parameters and clinical trial information. (a) Model parameters for the base model. (b) Patient-specific parameters for the in vivo model. (c) Clinical trial information in our simulation.

• MDAR checklist

### Data availability

The model code and source data are included in the GitHub https://github.com/zhoujw14/BiTE-Code.git (copy archived at *Zhou, 2022*).

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

# Appendix 1

## 1.1 Antibody-antigen binding to form binary complex

The following schema described antibody-antigen binding to form binary complex. Two antigens (CD3 (A) and CD19 (B)) competing for the same BiTE antibody (Y) was considered as a 3D binding process among free molecules, which was supposed to reach the equilibrium rapidly after co-culture initiation.

## A: CD3;   B: CD19;   Y: BiTE

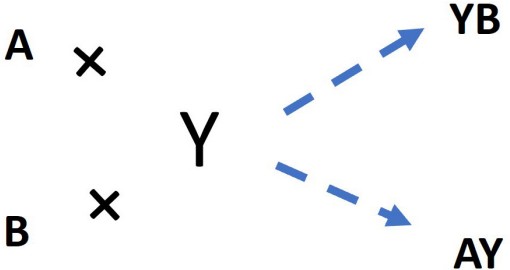

**Appendix 1—scheme 1.** The schema for antibody-antigen binding to form the binary complex.

The equations to solve free antigen ([A] and [B]) and binary complex ([AY] and [YB]) concentration levels at equilibrium (*Yan et al., 2012*):

- Total concentrations:

$$[A]_{tot} = [A] + [AY] \; ; \; [B]_{tot} = [B] + [YB] \; ; \; [Y]_{tot} = [Y] + [AY] + [YB] \tag{1}$$

- Dissociation constants:

$$K_{d,A} = [A] \cdot [Y] / [AY] \; ; \; K_{d,B} = [B] \cdot [Y] / [YB] \tag{2}$$

- Get binary complex concentrations ([AY] and [YB]) from *equation (1) and (2)*:

$$[AY] = \frac{[A] \cdot [Y]_{tot} / K_{d,A}}{1 + [A] / K_{d,A} + [B] / K_{d,B}} \; ; \; [YB] = \frac{[B] \cdot [Y]_{tot} / K_{d,B}}{1 + [A] / K_{d,A} + [B] / K_{d,B}} \tag{3}$$

- Where free antigen concentrations ([A] and [B]) at equilibrium (*Yan et al., 2012*):

$$[A] = \frac{[A]_{tot} \cdot K_{d,A}}{K_{d,A} + [Y]_{tot} \cdot (1 - z)} \; ; \; [B] = \frac{[B]_{tot} \cdot K_{d,B}}{K_{d,B} + [Y]_{tot} \cdot (1 - z)} \tag{4}$$

- Where z is the solution of a polynomial equation (for z solution see *Yan et al., 2012*):

$$\textit{If } K_{d,A} \neq K_{d,B}, \;\; \textit{the polynomial is cubic}: \;\; z^3 + bz^2 + cz + d = 0 \tag{5}$$

$$z \, satisfies, \; 0 < z < a_A + a_B \, and \, z < 1, \tag{6}$$

$$
\begin{aligned}
Here, \; a_A &= \frac{[A]_{tot}}{[Y]_{tot}}, \; a_B = \frac{[B]_{tot}}{[Y]_{tot}} \; , \; k_A = \frac{K_{d,A}}{[Y]_{tot}}, k_B = \frac{K_{d,B}}{[Y]_{tot}} \\
b &= -\left(2 + k_A + k_B + a_A + a_B\right) \\
c &= 1 + 2a_A + 2a_B + k_A + k_B + k_B a_A + k_A a_B + k_A k_B \\
d &= -\left(k_B a_A + k_A a_B + a_A + a_B\right)
\end{aligned}
$$

## 1.2 CD3 down-regulation and CD19 internalization within 1 h after binary-complex equilibrium

A: CD3; B: CD19; Y: BiTE

CD3 down-regulation (empirical equations):

$$[AY]_t^{'} = [AY]^{'} / \left(1 + 0.1 \cdot [AY]^{'\gamma} \cdot t^h\right) \tag{7}$$

$$[A]_t^{'} = [A]^{'} / \left(1 + 0.1 \cdot [AY]^{'\gamma} \cdot t^h\right) \tag{8}$$

where, for each effector cell, $[AY]^{'}$ and $[A]^{'}$ are assigned AY and A concentration on cell surface; $[AY]^{'}_t$ and $[A]^{'}_t$ are AY and A concentration on cell surface at time t; hill function: $\gamma$=0.9; h=0.7.

CD19 internalization:

$$[YB]_t^{'} = [YB]^{'} \cdot e^{-k_{int}t} \tag{9}$$

where, for each target cell, $[YB]^{'}$ is assigned YB concentration on cell surface; $[YB]^{'}_t$ is the YB concentration on cell surface at time t; $k_{int}$ stands for internalization rate constant (0.002 min⁻¹) (**Du et al., 2008**; **Ingle et al., 2008**)

The CD3 down-regulation over time at different blinatumomab concentrations was simulated as below.

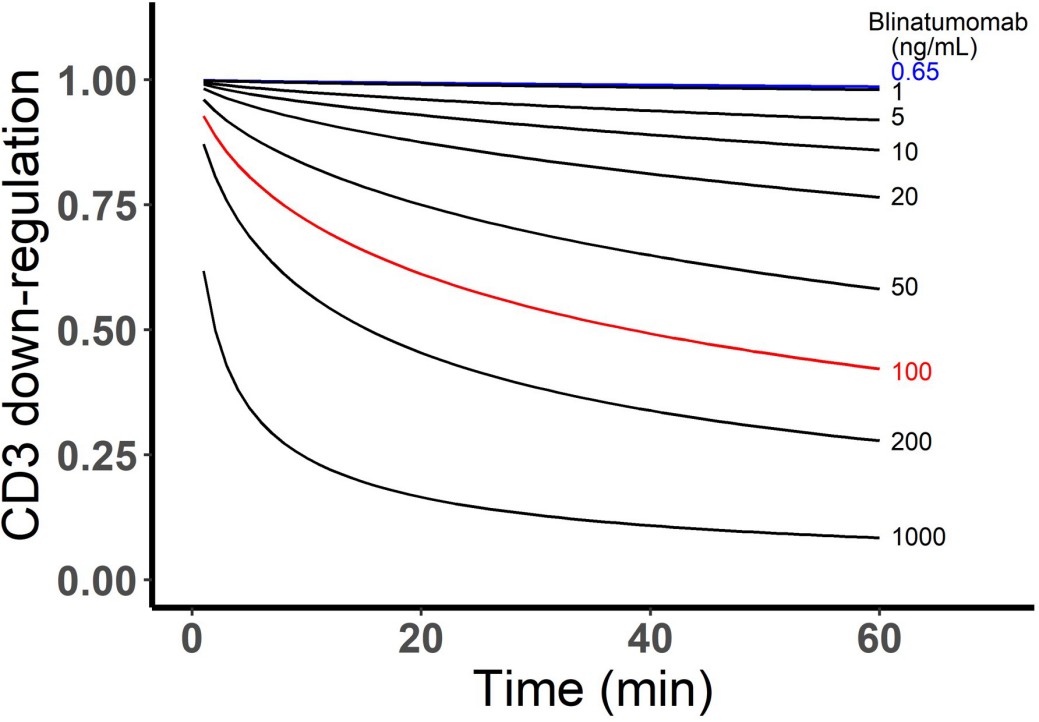

**Appendix 1—figure 1.** Simulated CD3 down-regulation over time at different blinatumomab concentrations. Initial setup: 2×106 total cells/mL, CD3 expression (L), CD19 expression (M), E:T ratio=1.

## 1.3 Equations and parameters for encounter probability

Encounter probability (**Celli et al., 2012**):

$$\text{for a single E encounter free T,} \quad P_e = 1 - e^{-\alpha Tt} \tag{10}$$

$$\text{for a ET encounter free E,} \quad P_{eE} = \left(1 - e^{-\alpha Et}\right) \cdot 2 \cdot f_{ETE} \tag{11}$$

$$\text{for a ET encounter free T,} \quad P_{eT} = \left(1 - e^{-\alpha Tt}\right) \cdot 2 \cdot f_{ETT} \tag{12}$$

$$\text{for a ETE encounter free E,} \quad P_{eEE} = \left(1 - e^{-\alpha Et}\right) \cdot 3 \cdot f_{ETEE} \tag{13}$$

$$\text{for a ETE encounter free T,} \quad P_{eET} = \left(1 - e^{-\alpha Tt}\right) \cdot 3 \cdot f_{ETET} \tag{14}$$

$$\text{for a ETT encounter free E,} \quad P_{eTE} = \left(1 - e^{-\alpha Et}\right) \cdot 3 \cdot f_{ETTE} \tag{15}$$

$$\text{for a ETT encounter free T,} \quad P_{eTT} = \left(1 - e^{-\alpha Tt}\right) \cdot 3 \cdot f_{ETTT} \tag{16}$$

$$\text{The } \alpha \text{ is given by,} \quad \frac{1}{\alpha} = \frac{R^3}{3Db} - \frac{3R^2}{5D} \tag{17}$$

where, Pe, encounter probability; E and T, cell number of free effector and target cells; t, time; f, spatial coefficients, the values were listed in *Supplementary file 1a*; α, mean hitting rate; D, cell diffusivity in solution, 0.83 µm²·s⁻¹ (*Miller et al., 2003*); b, distance between cell centers at encounter, 11 µm (radius: effector cell, 5 µm; target cell, 6 µm); R, radius of spherical co-culture system, 6200 µm (1 mL system).

The following schema depicted the spatial factor when a typical IS (ET) encounters an additional free target cell or effector cell. Based on imaging data, it was allowed a single effector cell to maximumly have 3 binding spots for target cells, whereas up to 4 effector cells for a single target cell. Dashed circles represent available spots for an additional effector or target cell binding to a typical IS (solid circles).

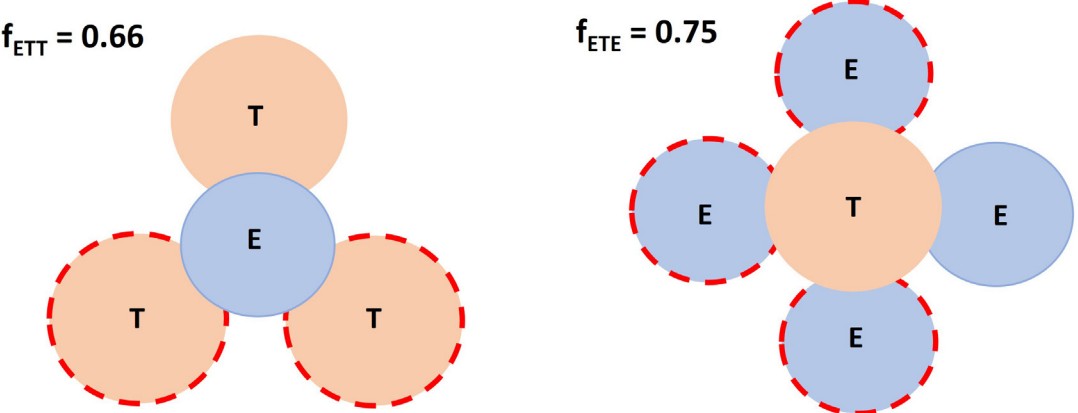

**f_ETT = 0.66**

**f_ETE = 0.75**

**Appendix 1—scheme 2.** Spatial factor when a typical immunological synapses (IS) (ET) encounters an additional free target cell or effector cell.

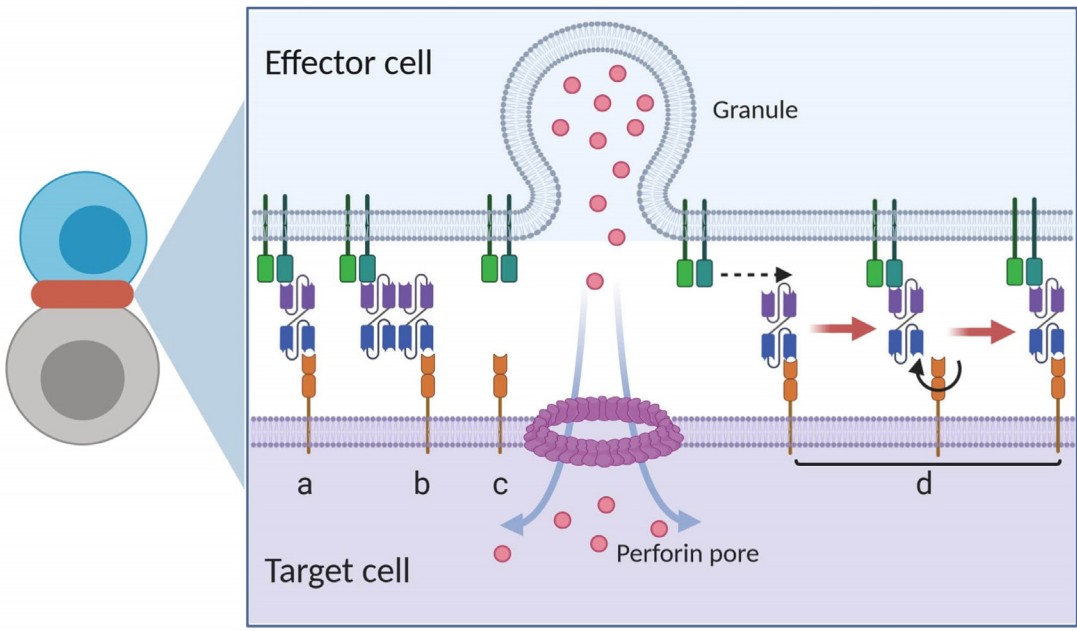

**Appendix 1—figure 2.** Graphical presentation of ternary complex (CD3-BiTE-CD19, bond) formation during cell-cell adhesion. The bond is formed by binding between a binary complex and a free antigen (a), rather than two binary complexes (b) or two free antigens (c). Three steps of 2-D binding on membrane (d): diffusion, rotation, and molecular binding.

## 1.4 Equations and parameters for ternary complex formation and adhesion probability

### A: CD3;   B: CD19;   Y: BiTE

$$A + YB \underset{k'_{off,\,A}}{\overset{k'_{on,\,A}}{\rightleftharpoons}} AYB \underset{k'_{off,\,B}}{\overset{k'_{on,\,B}}{\rightleftharpoons}} AY + B$$

**Appendix 1—scheme 3.** The formation of ternary complex (AYB, bond).

The formation of ternary complex (AYB, bond) can be characterized as

$$\frac{d\,[AYB]'}{dt} = k'_{on,\,A} \cdot [A]' \cdot [YB]' + k'_{on,\,B} \cdot [B]' \cdot [AY]' - k'_{off,\,A} \cdot [AYB]' - k'_{off,\,B} \cdot [AYB]' \tag{18}$$

$$\frac{d\,[A]'}{dt} = k'_{off,\,A} \cdot [AYB]' - k'_{on,\,A} \cdot [A]' \cdot [YB]' \tag{19}$$

$$\frac{d\,[AY]'}{dt} = k'_{off,\,B} \cdot [AYB]' - k'_{on,\,B} \cdot [B]' \cdot [AY]' \tag{20}$$

$$\frac{d\,[B]'}{dt} = k'_{off,\,B} \cdot [AYB]' - k'_{on,\,B} \cdot [B]' \cdot [AY]' \tag{21}$$

$$\frac{d\,[YB]'}{dt} = k'_{off,\,A} \cdot [AYB]' - k'_{on,\,A} \cdot [A]' \cdot [YB]' \tag{22}$$

Adhesion probability (**Chesla et al., 1998**; **Huang et al., 2010**)

$$N_{bond,\,\tau} = [AYB]'_{\tau} \cdot S_{contact} \tag{23}$$

$$Pa = 1 - e^{-\beta \cdot N_{bond,\tau}} \tag{24}$$

Where, $k'_{on}$, $k'_{off}$, effective rate constants for 2-D binding on membrane (see Appendix 1.5 and **Supplementary file 1a**):

$[A]'$, $[B]'$, $[AY]'$, $[YB]'$, $[AYB]'$, antigen density on cell surface;
$N_{bond,\tau}$, bond number at time $\tau$;
$[AYB]'_{\tau}$, bond density at contact duration $\tau$ (randomly assigned from 0.1 to 5 s);
$S_{contact}$, apparent contact area, ~5 μm²;
$Pa$, adhesion probability;
β, sensitive factor, 0.033 (for base and in-vitro model).

## 1.5 'Single-step model' for 2-D binding on cell membrane

## A: CD3;   B: CD19;   Y: BiTE

**Appendix 1—scheme 4.** 'Single-step model' for 2-D binding on cell membrane.

Equations for effective kinetic constants for 2-D binding on cell membrane:
• 3-D binding in solution (**Bell, 1978**; **Faro et al., 2017**)

$$d_+ = 4\pi \cdot R_{AYB} \cdot (D_{AY} + D_B) ; \qquad d_- = 3 \cdot R_{AYB}^{-2} \cdot (D_{AY} + D_B) \tag{25}$$

$$e_- = 0.04 \cdot e_- ; \qquad e_- = 3 \cdot d_-/4\pi^2 \tag{26}$$

$$k_{on} = \frac{d_+ \cdot e_+ \cdot r_+}{d_- \cdot e_- + r_+(d_- + e_+)} ; \qquad k_{off} = \frac{d_- \cdot e_- \cdot r_-}{d_- \cdot e_- + r_+(d_- + e_+)} \tag{27}$$

Similar equations were applied for A+BY ⇌ AYB as well
• 2-D binding on membrane (**Bell, 1978**; **Faro et al., 2017**)

$$d'_+ = 2\pi \cdot \left(D'_{AY} + D'_B\right) ; \qquad d'_- = 2 \cdot R_{AYB}^{-2} \cdot \left(D'_{AY} + D'_B\right) \tag{28}$$

$$e'_+ = 0.04 \cdot e'_- ; \qquad e'_- = 3 \cdot d'_-/4\pi^2 \tag{29}$$

$$k'_{on} = \frac{d'_+ \cdot e'_+ \cdot r_+}{d'_- \cdot e'_- + r_+ \left(d'_- + e'_+\right)} ; \qquad k'_{off} = \frac{d'_- \cdot e'_- \cdot r_-}{d'_- \cdot e'_- + r_+ \left(d'_- + e'_+\right)} \tag{30}$$

• Similar equations were applied for A+BY ⇌ AYB as well
Where, $d_+$, $d_-$, $d'_+$, $d'_-$, forward and reverse rate constants of antigen encounter;

$e_+, e_-, e'_+, e'_-$, forward and reverse rate constants of antigen rotation;

$r_+, r_-$, chemical association and dissociation rate constants;

$k_{on}, k_{off}, k'_{on}, k'_{off}$, effective rate constants, see *Supplementary file 1a*;

$D_{AY}, D_B$, diffusion constant in solution, $D_{AY} = D_B = D_Y$ , 50 μm²/s (*Bell, 1978*);

$D'_{AY}, D'_B$, diffusion constant on membrane, $D'_{AY} + D'_B$ = 0.01 μm²/s (*Bell, 1978*);

$R_{AYB}$, distance of reactants (AY and B), 5 nm (*Jansson, 2010*).

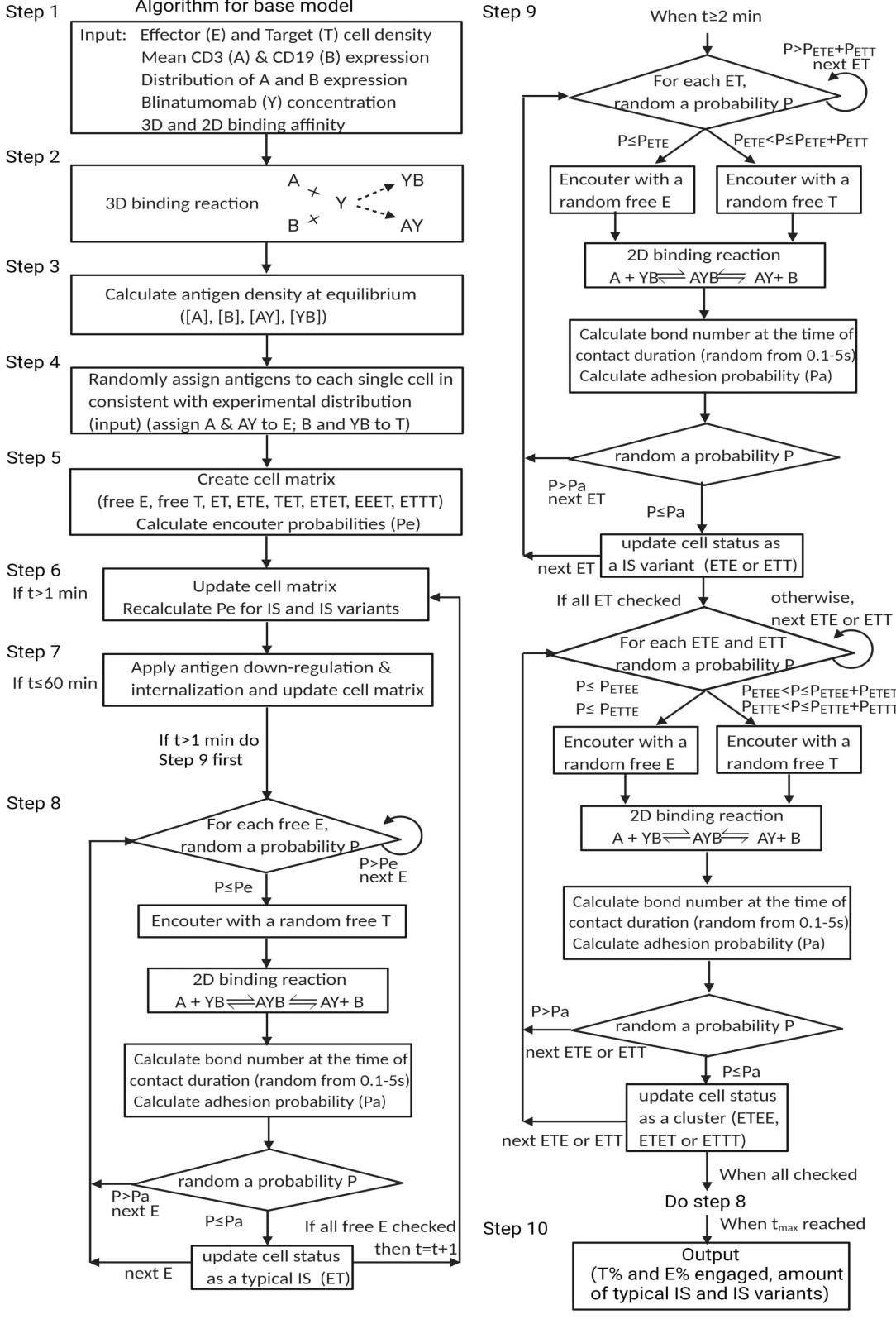

**Appendix 1—figure 3.** Algorithm for the base model.

