## [Editor Report]

This work provides an important finding, that aspects of clinical outcomes can be predicted by a random search to an immunological synapse-based computational model for T cells directed by specific engagers. It provides solid evidence based on in vitro synapse formation measurements using imaging flow cytometry. The work will be of interest to investigators in the still-expanding immunotherapy field, and also as an example of how biologic drugs interface with endogenous cellular resources in a patient.’

---

## [Decision Letter]

**Decision letter after peer review:**

Thank you for submitting your article "Population Dynamics of Immunological Synapse Formation Induced by Bispecific T-cell Engagers Predict Clinical Pharmacodynamics and Treatment Resistance" for consideration by *eLife*. Your article has been reviewed by 3 peer reviewers, including Michael L Dustin as the Reviewing Editor and Reviewer #1, and the evaluation has been overseen by Aleksandra Walczak as the Senior Editor. The following individual involved in review of your submission has agreed to reveal their identity: Xiling Jiang (Reviewer #2).

Essential revisions:

1. Cell adhesion molecules are regulated by TCR are not explicitly included in the model. Do the author feel that BITEs are not triggering these mechanisms or that their action is directly related to BiTE mediated interactions that are explicitly modelled such that the contributions of adhesion are included in the contributions of the BiTE interactions.

2. T cell movements in tissue is not a random walk, but is scaffolded by stromal cells and extracellular matrix to generate a variety of search strategies, which have been reported in the literature. Do any of the surprising results from the model potentially arise from known, non-Brownian aspects of T cell migration in tissues (or from aspects of cell adhesion in tissues that are not linear related to BiTE mediated interactions).

*Reviewer #1 (Recommendations for the authors):*

The authors use Jurkat-Raji combination for imaging flow cytometry. Jurkat is likely to use the CD2-CD58 adhesion system to engage Raji and this could be investigated with antibodies to CD2 or CD58. It is not clear to me how the simple use of blinatumomab bridge formation can correctly model the adhesion process. I suspect this works because the CD2-CD58 system is likely to be a constant and the frequency of blinatumomab Bridges is controlling signaling that promoted CD2-CD58 interaction to mediate conjugate formation and observed actin polymerisation. So the modelling may be correct in predicting the bell shape, but would not hold up if the physical requirements for synapse formation were correctly modelled. So the inclusion of CD2-CD58 and other adhesion systems may not change some aspects of the models, but it would provide another escape route for the tumour- through CD58 loss rather than CD19 loss.

Understanding the competition between adhesion and motility in the tumour may be important to understand the dissociation process. So chemokinesis may be an important thing to consider that may different in different settings. Due to chemokine like CXCL12 and CCL19/21 it may be very strong in lymphoid tissues, but bone marrow and other tissues may have very different landscapes of chemokine and thus the drive to disengage from a target may be lower or higher.

In terms of search models, the actual movement pattern will depend upon the underlying stroma. In a lymph node this network allows for frequent turns and exploration of the 3D space, where in some tumours the stroma may be more oriented and may convey T cells toward or away from the tumour. In the CNS, Lévy flight was found to be more efficient than a random walk would have been. Can this complexity explain any situation where the model predictions didn't hold?

*Reviewer #2 (Recommendations for the authors):*

1. What is the major purpose of modeling the immune synapse variants? Are they expected to have clinical significance (e.g., increase /decrease in efficacy, induce tumor antigen escape)?

2. In page 10 Figure 3, the incubation system was defined with X x 10E6 cells/mL, does this refer to total cell numbers (i.e., E + T cells), effector cells numbers or target cell numbers, please clarify in the figure legend for each respective experimental condition (e.g., Figure 3h and 3i, when you have different E:T ratios).

3. In page 10 Figure 3h, I would like recommend separation of effector cells and target cells, given that engagement of target cells is expected to be more clinically relevant.

4. In page 12 line 233, it was stated that "IS formation was optimized when the E:T ratio was around 1". Given that a lot of in vitro studies a conducted using E:T ratio 5:1 or 10:1, do you think your simulation results can be used to modify the in vitro study experimental condition towards better outcome?

5. Page 15 Figure 5c and 5d, do you have any observed data to verify the model simulated reduction in CD19 expression level following blinatumomab treatment?

6. Page 15 Figure 5e and 5f, one major concern for me is that your model simulation suggested that the bispecific is more effective in spleen and bone marrow compared to that in bone marrow, which generally against the clinical observation (e.g., the expected efficacious dose of blinatumomab for Acute Lymphoblastic Leukemia [major site of action is bone marrow] and Non-Hodgkin's Lymphoma [major site of action is lymph node] are 15 ug/m2/day vs. 60 ug/m2/day, respectively), and animal data (e.g., MGD-011 showed much strong B cell depletion effect in bone marrow compared to that in spleen and lymph node, PMID: 27663593). This may be associated with the heterogenous distribution of T cells and B cells in the lymph node and spleen (https://www.google.com/url?sa=iandurl=https%3A%2F%2Fimmunox.ucsf.edu%2Fsites%2Fimmunox.ucsf.edu%2Ffiles%2Fpdf%2FMicro204_Anat_IR%2520v2018.pdfandpsig=AOvVaw0qOMQka3SNN7k762B84FYDandust=1670808103053000andsource=imagesandcd=vfeandved=0CBAQjhxqFwoTCLC0za2z8PsCFQAAAAAdAAAAABAd). Please update your model simulation and associated context accordingly.

7. Page 17 Figure 6. Same issue as Figure 5e and 5f, with the model simulated regimen, we are not supposed to expect more B cell depletion in lymph node and spleen compared to that in bone marrow.

8. Page 20 Figure7. You may consider alternative regimens (e.g., high dose intensive treatment initially, followed by lower dose, less intensive treatment for consolidation) given that E:T ratio is expected to increase substantially following initial treatment.

9. Figure 5—figure supplement 3C. I'm not sure if the conclusion that "bidirectional effect was shown by increasing B cell density, owing to enhanced probability of cell-cell encounter and then insufficient BiTE concentration" hold true, given that for each individual B cell, the opportunity of encounter blinatumomab and T cells should be the same even at lower B cell concentrations.

10. Figure 4—figure supplement is missing

11. Given that you have 17 supplementary figures, please include a separate file where all the figures and the respective figure legends will be arranged together when you submit the revised article.

*Reviewer #3 (Recommendations for the authors):*

1. More details regarding the estimation of model parameters should be provided. Specifically, details of the type of the cost function used, confidence intervals, and the sensitivity of the in vivo dosage strategies to the chosen parameter values. It was not clear if a killing rate for tumors was used and whether it was estimated. Do the T cells proliferate/die in the in vivo model? A clear discussion of the data that were used to train the model (e.g., to estimate parameters) and test predictions should help.

2. It might help to further evaluate the importance of the cell population level interactions added in the model if one of the existing models in the literature was compared against the model developed here for describing the in vitro and in vivo experiments.

3. I think an experimental test of the surprising results in supplementary Figure 3a will substantially increase the confidence in the model.

4. It will help the readers to follow the model if model parameters were shown in the figures (e.g., Figure 5a).

---

## [Author Response]

Essential revisions:1. Cell adhesion molecules are regulated by TCR are not explicitly included in the model. Do the author feel that BITEs are not triggering these mechanisms or that their action is directly related to BiTE mediated interactions that are explicitly modelled such that the contributions of adhesion are included in the contributions of the BiTE interactions.

We agree with the reviewer that adhesion molecules play critical roles in synapse formation. In our model, we assumed these adhesion molecules were constant and comparable across cell populations. This assumption allowed us to focus on the BiTE-mediated interactions.

To clarify this point, we added a few sentences in the manuscript:

“Adhesion molecules such as CD2-CD58, integrins and selectins, are critical for cell-cell interaction. The model did not consider specific roles played by these adhesion molecules, which were assumed constant across cell populations. The model performed well under this simplifying assumption.”

In addition, we acknowledged the fact that “synapse formation is a set of precisely orchestrated molecular and cellular interactions. Our model merely investigated the components relevant to BiTE pharmacologic action and thereby serve as a simplified representation of this process”.

2. T cell movements in tissue is not a random walk, but is scaffolded by stromal cells and extracellular matrix to generate a variety of search strategies, which have been reported in the literature. Do any of the surprising results from the model potentially arise from known, non-Brownian aspects of T cell migration in tissues (or from aspects of cell adhesion in tissues that are not linear related to BiTE mediated interactions).

We agree that the tissue stromal factors greatly influence the patterns of T cell searching strategy. Our current model considered Brownian motion as a good first approximation for two reasons: (1) we define tissues as homogeneous compartments to attain unbiased evaluations of factors that influence BiTE-mediated cell-cell interaction, such as T cell infiltration, T: B ratio, and target expression. The stromal factors were not considered in the model, as they require spatially resolved tissue compartments to represent the gradients of stromal factors; (2) our model was primarily calibrated against in vitro data obtained from a “well-mixed” system that does not recapitulate specific considerations of tissue stromal factors. We did not obtain tissue-specific data to support the prediction of T cell movement. This is under current investigation in our lab. Therefore, we are cautious about assuming different patterns of T cell movement in the model when translating into in vivo settings. We acknowledged the limitation of our model for not considering the more physiologically relevant T-cell searching strategies.

In the Discussion, we added a limitation of our model: “We assumed Brownian motion in the model as a good first approximation of T cell movement. However, T cells often take other more physiologically relevant searching strategies closely associated with many stromal factors. Because of these stromal factors, the cell-cell encounter probabilities would differ across anatomical sites.”

Reviewer #1 (Recommendations for the authors):The authors use Jurkat-Raji combination for imaging flow cytometry. Jurkat is likely to use the CD2-CD58 adhesion system to engage Raji and this could be investigated with antibodies to CD2 or CD58. It is not clear to me how the simple use of blinatumomab bridge formation can correctly model the adhesion process. I suspect this works because the CD2-CD58 system is likely to be a constant and the frequency of blinatumomab Bridges is controlling signaling that promoted CD2-CD58 interaction to mediate conjugate formation and observed actin polymerisation. So the modelling may be correct in predicting the bell shape, but would not hold up if the physical requirements for synapse formation were correctly modelled. So the inclusion of CD2-CD58 and other adhesion systems may not change some aspects of the models, but it would provide another escape route for the tumour- through CD58 loss rather than CD19 loss.

We appreciate the suggestions on the adhesion model and motility pattern. Adhesion is a very complex process containing a lot of signaling and functional proteins. We agree that the CD2-CD58 adhesion is critical for cell-cell interaction. Unfortunately, CD58 expression across cell populations was not quantified in our experiments and our model assumed these adhesion molecules stay constant as a part of the BiTE-mediated interactions. In our model, the adhesion probability is approximated as a function of trinary complexes (BiTE bridging). Such approximation significantly reduced the computational complexity but retain sufficient accuracy. The stochastic nature of the model does account for the influence of other molecules, like CD2-CD58 interaction. However, we did not explicitly put these molecules into our model for quantitative evaluation. We have stated this within the text as a model assumption and limitation.

We added the following contents into our manuscript. “Adhesion molecules such as CD2-CD58, integrins and selectins, are critical for cell-cell interaction. The model did not consider specific roles played by these adhesion molecules, which were assumed constant across cell populations. The model performed well under this simplifying assumption.”

Understanding the competition between adhesion and motility in the tumour may be important to understand the dissociation process. So chemokinesis may be an important thing to consider that may different in different settings. Due to chemokine like CXCL12 and CCL19/21 it may be very strong in lymphoid tissues, but bone marrow and other tissues may have very different landscapes of chemokine and thus the drive to disengage from a target may be lower or higher.

This is a great point. Although our model did not consider chemokines like CXCL12 and CCL19/21, these chemokine gradients likely alter the kinetics of engagement/disengagement in a tissue-specific manner. As demonstrated, differences in T cell infiltration and T: B ratios across anatomical sites create sanctuary sites for tumor cells in the host. The different landscapes of these chemokines across tissues would make certain sanctuary sites, like the bone marrow, more likely to manifest.

We have added the following content to the discussion. “The inadequate tumor lysis in the bone marrow might also be explained by tissue-specific differences chemokine gradients that hinder cell-cell interaction and adhesion.”

In terms of search models, the actual movement pattern will depend upon the underlying stroma. In a lymph node this network allows for frequent turns and exploration of the 3D space, where in some tumours the stroma may be more oriented and may convey T cells toward or away from the tumour. In the CNS, Lévy flight was found to be more efficient than a random walk would have been. Can this complexity explain any situation where the model predictions didn't hold?

We agree that T cell motility is quite different between in vitro and in vivo settings, as well as across anatomical sites. Random search is merely an approximation of T cell behaviors in our model and we agree that tissue stromal factors greatly influence the patterns of T cell searching strategy. Our current model considered Brownian motion as a good first approximation for two reasons: (1) we define tissues as homogeneous compartments to attain unbiased evaluations of factors that influence BiTE-mediated cell-cell interaction, such as T cell infiltration, T: B ratio, and target expression. Stromal factors were not considered in the model, as they often need spatially resolved tissue model to accommodate the gradients of these stromal factors; (2) our model was primarily calibrated against in vitro data obtained from a system without tissue stromal factors, and we did not obtain tissue-specific data in this study to support the predictions of T cell movement patterns influenced by these stromal factors. This is under current investigation in our lab. Therefore, we are cautious about assuming different T cell movement patterns in the model when translating BiTE-mediated cell-cell interactions from in vitro to in vivo settings. We acknowledged the limitations of our model for not considering the more physiologically relevant T-cell searching strategies.

Reviewer #2 (Recommendations for the authors):1. What is the major purpose of modeling the immune synapse variants? Are they expected to have clinical significance (e.g., increase /decrease in efficacy, induce tumor antigen escape)?

This is a great point. Immune synapse variants account for 9.7 ~ 50.1% of total IS, depending on the experimental conditions. These variants cannot be ignored, especially at high cell density and high E: T ratio (Figure 4a, condition 5 and 11). It remains unclear how frequently cell clustering occurs in vivo, as well as what drives it; we speculate that cell clustering (e.g., ETEE) may yield a higher killing efficiency as shown in an efficacy study (Gong et al. 2019). This aligns intuitively with higher E:T ratios being associated with greater tumor killing in both in vitro and in vivo systems; there is simply a greater density of effector cells in the system, increasing the likelihood of interaction through Brownian motion. Our model does not differentiate the killing efficiency of these synapse variants, but the model does provide a framework for us to address these questions in the future.

2. In page 10 Figure 3, the incubation system was defined with X x 10E6 cells/mL, does this refer to total cell numbers (i.e., E + T cells), effector cells numbers or target cell numbers, please clarify in the figure legend for each respective experimental condition (e.g., Figure 3h and 3i, when you have different E:T ratios).

It refers to total cell numbers (E+T cells).

We have added this clarification in the figure legend, like “2X, 2x10^6 total cells/mL”

3. In page 10 Figure 3h, I would like recommend separation of effector cells and target cells, given that engagement of target cells is expected to be more clinically relevant.

Thank you. Figure 3i and the original Figure 3-supplement 1 show separated curves for the percentages of effector cells and target cells engaged in immunological synapses. We have removed Figure 3h and replaced by original Figure 3-supplement 1.

4. In page 12 line 233, it was stated that "IS formation was optimized when the E:T ratio was around 1". Given that a lot of in vitro studies a conducted using E:T ratio 5:1 or 10:1, do you think your simulation results can be used to modify the in vitro study experimental condition towards better outcome?

A great point, it was misleading as shown in our original Figure 3h and the above statement.

To avoid confusion, we have removed Figure 3h and replaced by original Figure 3-supplement 1. To provide an explanation, “E:T ratio around 1”, is the optimized condition that yielded the highest fraction of total cells engaged in an IS, not the fraction of target cells. We agree that it is not a good indicator of clinical outcome. In contrast, the fraction of target cells engaged is expected to be lower at 1:1 ET ratio than at a higher E: T ratio (Figure 3i). Using higher E:T ratios (e.g., 5:1 or 10:1) would clearly maximize killing efficacy.

5. Page 15 Figure 5c and 5d, do you have any observed data to verify the model simulated reduction in CD19 expression level following blinatumomab treatment?

We did not experimentally verify CD19 evolution in our in vitro system, as the timescale for immunological synapse formation was too short to expect adaptation and CD19 negative “relapse” (1 hr). Considering T-cells are serial killers of tumor cells, we did not extend our observation to a longer duration as it would prevent an accurate quantification of immunological synapses. However, antigen escape has been a tumor resistance mechanism broadly observed in the literature (Xu et al., 2019; Mejstríková et al., 2017; Samur et al., 2021).

We have cited these references and meanwhile acknowledged that more validations of our models are warranted.

6. Page 15 Figure 5e and 5f, one major concern for me is that your model simulation suggested that the bispecific is more effective in spleen and bone marrow compared to that in bone marrow, which generally against the clinical observation (e.g., the expected efficacious dose of blinatumomab for Acute Lymphoblastic Leukemia [major site of action is bone marrow] and Non-Hodgkin's Lymphoma [major site of action is lymph node] are 15 ug/m2/day vs. 60 ug/m2/day, respectively), and animal data (e.g., MGD-011 showed much strong B cell depletion effect in bone marrow compared to that in spleen and lymph node, PMID: 27663593). This may be associated with the heterogenous distribution of T cells and B cells in the lymph node and spleen (https://www.google.com/url?sa=iandurl=https%3A%2F%2Fimmunox.ucsf.edu%2Fsites%2Fimmunox.ucsf.edu%2Ffiles%2Fpdf%2FMicro204_Anat_IR%2520v2018.pdfandpsig=AOvVaw0qOMQka3SNN7k762B84FYDandust=1670808103053000andsource=imagesandcd=vfeandved=0CBAQjhxqFwoTCLC0za2z8PsCFQAAAAAdAAAAABAd). Please update your model simulation and associated context accordingly.

Thank you for providing this information. It is very helpful for us to justify our model assumptions. We agree that the heterogeneous distribution of T and B cells can reduce drug efficacy. Meanwhile, as shown in Reviewer 1 comments, the bone marrow has very different chemokines and a dense stromal structure, which may reduce T cell mobility and functionality. Because of these aspects, the bone marrow often has reduced drug efficacy, consistent with our model prediction. However, it remains to be determined which organs restrict drug efficacy to the highest degree. We suggest that the bone marrow could be an organ that contributes to tumor evolution, consistent with literature showing that tumor relapses are often firstly detected in the bone marrow. One recent paper published in PLoS Com Bio also showed reduced efficacy in the bone marrow (Yoneyama et al., 2022). This is an issue for further clarification and confirmation.

To acknowledge the inconsistent observations, we added the following content, “Our models only considered a few select factors that influence the formation of IS, which may not provide a full description of drug inhomogeneous efficacy across anatomical sites. Factors like the heterogeneous distribution of T and B cells, chemokines, and stromal structures could affect the T cell motility and functions in tissue environments and including these factors may provide an unbiased evaluation of drug effect across tissues.”

7. Page 17 Figure 6. Same issue as Figure 5e and 5f, with the model simulated regimen, we are not supposed to expect more B cell depletion in lymph node and spleen compared to that in bone marrow.

Thank you for this point. We have made some adjustment in our manuscript and details of the explanation is provided in the last question.

8. Page 20 Figure7. You may consider alternative regimens (e.g., high dose intensive treatment initially, followed by lower dose, less intensive treatment for consolidation) given that E:T ratio is expected to increase substantially following initial treatment.

**:** In agreement with reviewer’s suggestion, we simulated alternative dosing regimens with high dose followed by medium dose and medium dose followed by low dose (see Author response image 1). At the given scenario, the alternative dosing regimen (dashed lines) does not perform as well as its counterpart dosing regimen (constant dose levels, solid lines). Although E:T ratio is expected to be high following the initial high doses in the first 2 cycles, the change of medium dose (dashed blue line) provides even less effect compared with constant medium dose (solid red line) since week 12. This is attributed to more CD19 loss after initial high dose compared with constant medium dose. Therefore, alternative regimen is not supported by our current model assumptions to maintain a comparable killing effect with constant high or medium doses.

**Author response image 1. sa2fig1:** 

9. Figure 5—figure supplement 3C. I'm not sure if the conclusion that "bidirectional effect was shown by increasing B cell density, owing to enhanced probability of cell-cell encounter and then insufficient BiTE concentration" hold true, given that for each individual B cell, the opportunity of encounter blinatumomab and T cells should be the same even at lower B cell concentrations.

We agree the expression is not clear. “Probability of cell-cell encounter” in the text actually indicates the overall probability for the population (T and B cells) not the individual B cell. When the T cell density is constant, the probability for each B cell encountering a T cell is independent of B cell density. At the population level, however, cell-cell encounter probability depends on both T cell and B cell density. At a higher B cell density, the probability for each T cell encountering a B cell stays high.

To avoid confusion, we revised this conclusion as “bidirectional effect was shown by increasing B cell density, owing to enhanced probability of cell-cell encounter at the population level and then insufficient BiTE concentration.”

10. Figure 4—figure supplement is missing

There is no supplement for figure 4, and Figure legends have been added to avoid confusion.

11. Given that you have 17 supplementary figures, please include a separate file where all the figures and the respective figure legends will be arranged together when you submit the revised article.

This is a good point. A separate arrangement of the supplementary figures and tables (Supplementary Materials) will be reviewer friendly. We have made the adjustments.

Reviewer #3 (Recommendations for the authors):1. More details regarding the estimation of model parameters should be provided. Specifically, details of the type of the cost function used, confidence intervals, and the sensitivity of the in vivo dosage strategies to the chosen parameter values. It was not clear if a killing rate for tumors was used and whether it was estimated. Do the T cells proliferate/die in the in vivo model? A clear discussion of the data that were used to train the model (e.g., to estimate parameters) and test predictions should help.

We did not perform a global parameter optimization of our models considering the stochastic agent-based nature of the models. However, the majority of key parameters was obtained or derived from the literature, such as 3D dissociation constant (supplementary file 1a), 2D on/off rate constant (derived, supplementary file 1a and Appendix 1.5), encounter probability (supplementary file 1a, Appendix 1.3), and CD19 internalization (supplementary file 1a). CD3 and CD19 distributions on cell surface (supplementary file 1a) were exactly aligned with our observation (Figure 1—figure supplement 1). IS variant formation and spatial coefficient were based on our observation and derived from spatial configuration (supplementary file 1a, Appendix 1.3). The data from our pilot studies were used to estimate the parameters in CD3 down-regulation (r and h in supplementary file 1a) and adhesion probability (β in supplementary file 1a) in our base model. Our in-vivo model for clinical prediction adopted the same parameter values from the base model and the in-vitro model. Newly added patient-specific parameters such as cell density in organs, cell turnover rate, cell trafficking rate, antigen expression, and drug distribution were all from literatures and clinical reports (supplementary file 1b and c). The only parameter we manually optimized in the in-vivo model is the sensitive coefficient (β in supplementary file 1b) for cell-cell adhesion and the value was re-calibrated against the in-vitro data at a low BiTE concentration. BiTE concentrations in patients (mostly < 2 ng/ml) is only relevant to the low bound of the concentration range we investigated in vitro (0.65-2000 ng/ml).

About T cells proliferate/die and killing rate, the response has been provided previously.

We added a discussion of our model limitation to clarify these points:

“The majority of model parameters were obtained or derived from the literature, and we did not perform model optimization to get the optimal values of model parameters. The only parameter we manually optimized is the sensitive coefficient for cell-cell adhesion in the base and in-vivo model and the values were calibrated against the in-vitro data. Implementing model optimization algorithms would improve the predictability of the models.”

2. It might help to further evaluate the importance of the cell population level interactions added in the model if one of the existing models in the literature was compared against the model developed here for describing the in vitro and in vivo experiments.

This is an interesting point, but it is challenging to make direct comparison of our models with literature models because of different methods and model objectives. We have cited most if not all literature models. Our model focused on understanding the factors driving IS formation and tested if these driving factors play important roles for BiTE clinical pharmacodynamics and tumor evolution. Literature models were mostly developed to describe the concentration-dependent cytotoxic effect of BiTE antibodies. All models predicted bell-shaped relationships and concentration dependency. Our models yield additional insights concerning IS synapse formation dynamics, tumor evolution, and non-homogeneous effects across anatomical sites. These insights could benefit our understanding of clinical responses and optimal doses.

3. I think an experimental test of the surprising results in supplementary Figure 3a will substantially increase the confidence in the model.

Thank you for pointing this out. Supplementary Figure 3a illustrates the influence of CD3 affinity on the fraction of effector cells engaged in IS formation. High CD3 affinity counterintuitively decreases the fraction of effector cells in IS formation due to (1) rapid CD3 downregulation upon BiTE engagement; (2) more BiTE antibodies consumed per IS because of the high affinity. In addition, the influence of CD3 affinity on cytotoxic effect is not quite consistent in the literature and there are papers reporting that high CD3 affinity may result negative effect on BiTE cytotoxic effects (Chen et al., 2021; Dang et al., 2021). We have cited these papers in the manuscript and believe they support our predictions.

4. It will help the readers to follow the model if model parameters were shown in the figures (e.g., Figure 5a).

Thank you for the suggestion, we have added IS duration (150 min), encounter probability (Pe), and adhesion probability (Pa) to Figure 5a; B cell turnover rate (k_B cell_), and B cell trafficking rate (k_traff, in_ and k_traff, out_) to Figure 6a.

References:

Chen W, Yang F, Wang C, Narula J, Pascua E, Ni I, Ding S, Deng X, Chu ML, Pham A, Jiang X, Lindquist KC, Doonan PJ, Blarcom TV, Yeung YA, Chaparro-Riggers J. 2021. One size does not fit all: navigating the multi-dimensional space to optimize T-cell engaging protein therapeutics. *MAbs* 13:1871171. DOI: 10.1080/19420862.2020.1871171, PMID: 33557687

Dang K, Castello G, Clarke SC, Li Y, AartiBalasubramani A, Boudreau A, Davison L, Harris KE, Pham D, Sankaran P, Ugamraj HS, Deng R, Kwek S, Starzinski A, Iyer S, Schooten WV, Schellenberger U, Sun W, Trinklein ND, Buelow R, Buelow B, Fong L, Dalvi P. 2021. Attenuating CD3 affinity in a PSMAxCD3 bispecific antibody enables killing of prostate tumor cells with reduced cytokine release. *Journal for ImmunoTherapy of Cancer* 9:e002488. DOI: 10.1136/jitc-2021-002488, PMID: 34088740

Gong C, Anders RA, Zhu Q, Taube JM, Green B, Cheng W, Bartelink IH, Vicini P, Wang BPopel AS. 2019. Quantitative Characterization of CD8^+^ T Cell Clustering and Spatial Heterogeneity in Solid Tumors. *Frontiers in* Oncology 8:649. DOI: 10.3389/fonc.2018.00649, PMID: 30666298

Mejstríková E, Hrusak O, Borowitz MJ, Whitlock JA, Brethon B, Trippett TM, Zugmaier G, Gore L, Stackelberg AV, Locatelli F. 2017. CD19-negative relapse of pediatric B-cell precursor acute lymphoblastic leukemia following blinatumomab treatment. *Blood Cancer Journal* 7: 659. DOI: 10.1038/s41408-017-0023-x, PMID: 29259173

Samur MK, Fulciniti M, Samur AA, Bazarbachi AH, Tai YT, Prabhala R, Alonso A, Sperling AS, Campbell T, Petrocca F, Hege K, Kaiser S, Loiseau HA, Anderson KC, Munshi NC. 2021. Biallelic loss of BCMA as a resistance mechanism to CAR T cell therapy in a patient with multiple myeloma. *Nature Communications* 12:868. DOI: 10.1038/s41467-021-21177-5, PMID: 33558511

Xu X, Sun Q, Liang X, Chen Z, Zhang X, Zhou X, Li M, Tu H, Liu Y, Tu S, Li Y. 2019. Mechanisms of relapse after CD19 CAR T-cell therapy for acute lymphoblastic leukemia and its prevention and treatment strategies. *Frontiers in Immunology* 10:2664. DOI: 10.3389/fimmu.2019.02664, PMID: 31798590

Yoneyama T, Kim MS, Piatkov K, Wang H, Zhu AZX. 2022. Leveraging a physiologically-based quantitative translational modeling platform for designing B cell maturation antigen-targeting bispecific T cell engagers for treatment of multiple myeloma. *PLOS Computational Biology* 18: e1009715. DOI: 10.1371/journal.pcbi.1009715, PMID: 35839267